# Evolutionary trajectory of pattern recognition receptors in plants

Bruno Pok Man Ngou [1], Michele Wyler [2], Marc W. Schmid [2], Yasuhiro Kadota [1] ✉ & Ken Shirasu [1] ✉

Cell-surface receptors play pivotal roles in many biological processes, including immunity, development, and reproduction, across diverse organisms. How cell-surface receptors evolve to become specialised in different biological processes remains elusive. To shed light on the immune-specificity of cell-surface receptors, we analyzed more than 200,000 genes encoding cell-surface receptors from 350 genomes and traced the evolutionary origin of immune-specific leucine-rich repeat receptor-like proteins (LRR-RLPs) in plants. Surprisingly, we discovered that the motifs crucial for co-receptor interaction in LRR-RLPs are closely related to those of the LRR-receptor-like kinase (RLK) subgroup Xb, which perceives phytohormones and primarily governs growth and development. Functional characterisation further reveals that LRR-RLPs initiate immune responses through their juxtamembrane and transmembrane regions, while LRR-RLK-Xb members regulate development through their cytosolic kinase domains. Our data suggest that the cell-surface receptors involved in immunity and development share a common origin. After diversification, their ectodomains, juxtamembrane, transmembrane, and cytosolic regions have either diversified or stabilised to recognise diverse ligands and activate differential downstream responses. Our work reveals a mechanism by which plants evolve to perceive diverse signals to activate the appropriate responses in a rapidly changing environment.

Cell-surface receptors allow organisms to detect environmental changes. Plant cell-surface receptors play a crucial role in perceiving both self- and non-self-molecules, such as peptides, small proteins, lipids, and polysaccharides[1,2]. These receptors can be categorised as either receptor-like proteins (RLPs), which possess a transmembrane domain (TM), or receptor-like kinases (RLKs), which have both TM and kinase domains (KDs). The family sizes of RLPs and RLKs vary among plant species, and are believed to expand over evolutionary time in response to pathogen pressure[3,4]. Recognition of pathogen-associated molecular patterns (PAMPs) via pattern-recognition receptors (PRRs) is a fundamental mechanism (known as PAMP/

PRR-triggered immunity; PTI) which allows organisms to detect the presence of pathogens[1,5]. PAMP perception is conserved in animals, plants, fungi, and other eukaryotes[6–8]. Previous work has explored the origin and evolutionary trajectory of RLKs[9,10]. However, the origins of PRR families involved in PAMP perception in plants remain largely unclear.

While many cell-surface receptors in plants are involved in pathogen recognition, many with similar domain architectures are also engaged in other biological processes (Supplementary Fig. 1; Supplementary Note 1). Some of these receptors recognise self-molecules, such as phytohormones and phytocytokines, to regulate develop-

[1]RIKEN Center for Sustainable Resource Science, Yokohama, Japan. [2]MWSchmid GmbH, Glarus, Switzerland. ✉e-mail: yasuhiro.kadota@riken.jp; ken.shirasu@riken.jp

mental and reproductive processes[11,12] (Supplementary Fig. 1). Given the striking resemblance in their domain architecture, it is reasonable to infer that immunity- and development-related cell-surface receptors share a common origin. However, the evolutionary trajectory that led to their divergence and specialisation in distinct biological processes remains poorly understood.

## Results

### The origin and expansion of cell-surface receptors in the plant lineage

Plant cell-surface receptors that are known to participate in immunity, development, and reproductive processes include the LRR-, G-lectin-, Wall-associated kinase (WAK)-, Domain of Unknown Function 26

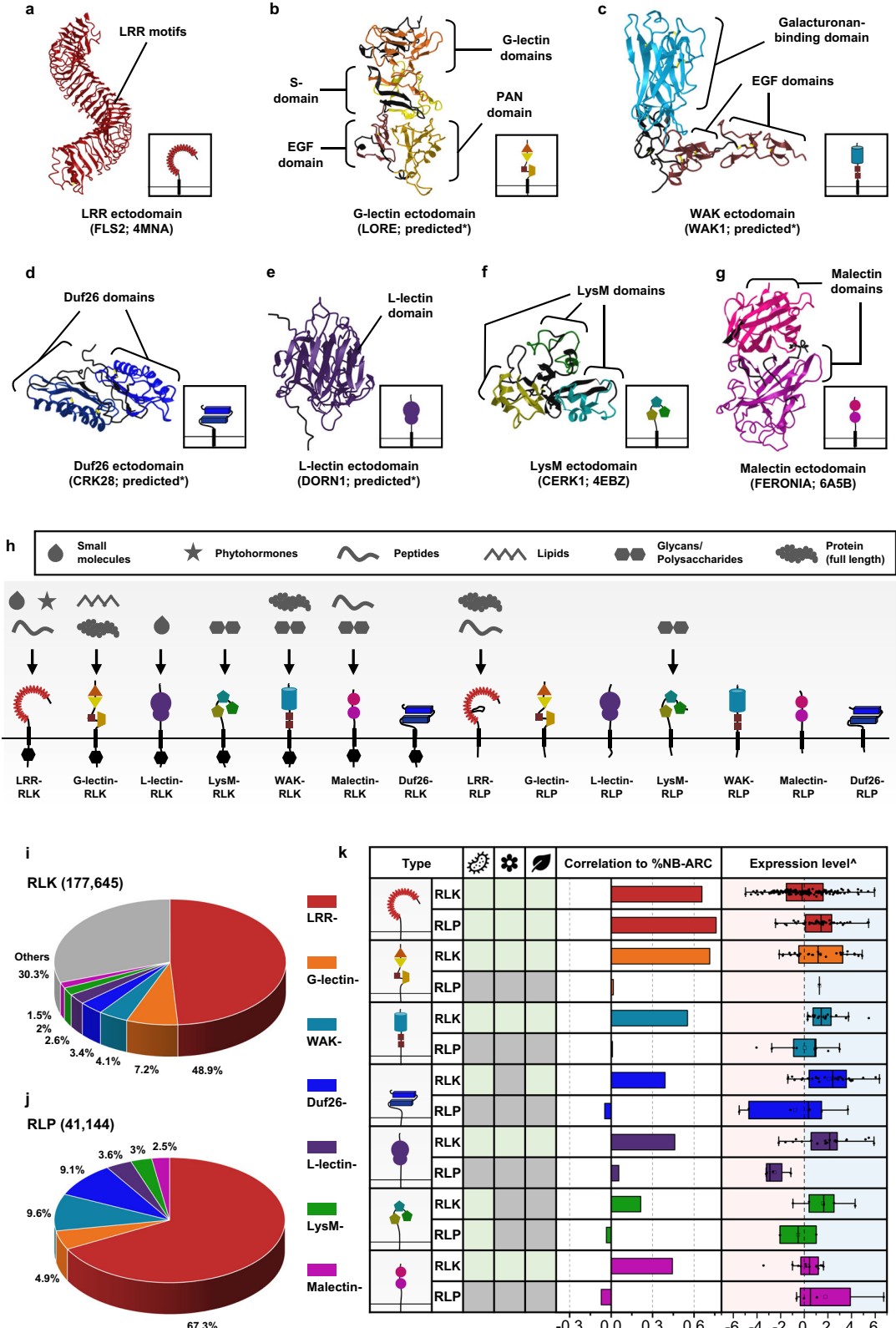

**Fig. 1 | The distribution of cell-surface receptors in plants. a–g** Ectodomain structure of an LRR receptor. **a** a G-lectin receptor, **b** an L-lectin receptor, **c** a LysM receptor, **d** a Malectin receptor, **e** a WAK receptor, **f** and a Duf26 receptor **g**. Structures of FLS2, CERK1, and FERONIA were published[55,95,96]. Structures of LORE, DORN1, WAK1 and CRK28 were predicted by Alphafold2*[97]. Ectodomains are visualized in iCn3D[98]. **h** Schematic displays the domain architecture of different classes of receptor-like kinase (RLKs) and receptor-like proteins (RLP) in plants. Arrows represent the ligands that these receptor classes have been reported to perceive or recognize. The upper box defines the ligands recognized by different receptors. The lower box defines the domains within the receptor classes. Note that these receptors may be able to recognise other unidentified ligands. For more information, see Supplementary Fig. 1. **i** Ectodomain distribution of RLKs in plants. Each fraction represents the percentage (%) of ectodomain out of all the RLKs from 350 species (177,645). **j** Ectodomain distribution of RLP in plants. Each fraction represents the percentage (%) of ectodomains out of all the RLPs with those seven ectodomains (41,144). **k** Table of RLKs and RLPs with LRR (red), G-lectin (orange), WAK (turquoise), Duf26 (blue), L-lectin (purple), LysM (green), and Malectin (magenta) ectodomains. Characterised receptors involved in microbial interaction (bacteria icon), reproduction (flower icon), and development (leaf icon) are indicated with light green boxes. Grey boxes indicate that the receptor class has not been reported to be involved in that biological process. For details, refer to Supplementary Fig. 1. Correlations between different classes of cell-surface receptors and NB-ARC in 300 angiosperms are indicated with bars. Strong positive correlations are indicated by extension to the light green area (Pearson's r > 0.6) and medium positive correlations are within the yellow area (Pearson's r between 0.3 and 0.6). Expression level^ refers to the expression of each class of cell-surface receptors during NLR-triggered immunity (NTI) in *Arabidopsis thaliana*. Light blue area represents increased expression and light pink area represents decreased expression during NTI. X-axis values represent $\log_2$ (fold change during ETI relative to untreated samples). Boxplot elements: center line, median; bounds of box, 25th and 75th percentiles; whiskers, 1.5 × IQR from 25th and 75th percentiles. Number of cell-surface receptors (n) analysed in the RNA-seq data: LRR-RLK, $n = 159$; LRR-RLP, $n = 42$; G-lectin-RLK, $n = 29$; G-lectin-RLP, $n = 1$; WAK-RLK, $n = 18$; WAK-RLP, $n = 10$; Duf26-RLK, $n = 33$; Duf26-RLP, $n = 7$; L-lectin-RLK, $n = 21$; L-lectin-RLP, $n = 4$; LysM-RLK, $n = 5$; LysM-RLP, $n = 2$; Malectin-RLK, $n = 13$; Malectin-RLP, $n = 4$. RNA-seq data analysed here were reported previously, where NTI was activated by estradiol-induced expression of AvrRps4 in *A. thaliana* for 4 h[94]. For the expression of each class of cell-surface receptors during PTI in *A. thaliana*, refer to Supplementary Fig. 2.

(Duf26)-, L-lectin-, Lysin motif (LysM)-, and Malectin-containing RLKs and RLPs (Fig. 1a–h). There are additional RLK families with different ectodomains, such as the proline-rich extensin-like receptor kinases (PERKs) and thaumatin-like protein kinases (TLPKs)[9,13]. However, their function in immunity is not well-characterized. Cell-surface receptors with LRR-, G-lectin-, WAK-, and LysM-ectodomains have been reported to recognise PAMPs, while others perceive self-molecules or unidentified ligands (Fig. 1h; Supplementary Fig. 1). Recognition of the diverse array of ligands is likely to be accomplished by variable structures and combinations of different ectodomains (Fig. 1a–g). To trace the origins of different receptor classes within the plant lineage, we first identified RLKs and RLPs in 350 genomes from Glaucophyta, red algae, green algae, Bryophytes, and Tracheophytes. We define here RLKs as any proteins with both 1–2 TMs and KDs, and RLPs as any protein with 1–2 TMs, but lack KDs. In total, we identified 177,645 RLKs, almost up to 70% of which possess either LRR-, G-lectin-, WAK-, Duf26-, L-lectin-, LysM- and Malectin-ectodomains (Fig. 1i). Next, we searched for proteins with these ectodomains and TMs that lack KDs and found 41,144 RLPs (Fig. 1j). We further examined which of the identified RLKs and RLPs families are likely to be involved in immunity. A previous report suggested a positive correlation between the gene family sizes of cell-surface immune receptors and intracellular immune receptors (the NB-ARC family) across the angiosperms[4]. We examined the correlation between the relative size (%; number of identified genes in the family/ numbers of searched genes × 100; see methods) of the RLK families, the RLPs families, and the NB-ARC family in each genome. Notably, most RLK families (except for the LysM-RLKs) exhibit positive correlations with the NB-ARC family, while most RLP families (except for the LRR-RLPs) do not exhibit positive correlation with the NB-ARC family (Main Fig. 1k). Furthermore, we checked the expression level of these receptor families in *Arabidopsis thaliana* during immunity. Notably, the RLKs, except for LRR- and Malectin-RLKs, generally exhibit higher expression levels compared to the RLPs during immunity (Main Fig. 1k; Supplementary Fig. 2). These data collectively suggest that the RLKs are more likely to be involved in immunity than the RLPs.

Next, we examined the presence or absence of ectodomains (LRR-, G-lectin-, WAK-, Duf26-, L-lectin-, LysM- and Malectin-ectodomains lacking TM or KD; ectodomain-only proteins), RLPs (TM-bound ectodomains) and RLKs (ectodomains encompassing both TM and KD) in the plant lineage (Fig. 2; Supplementary Fig. 3; Supplementary Data 1a–c). Ectodomains exhibit an ancient heritage, with LRR-, WAK-, LysM-, Malectin-, and L-lectin-domains dating back to the era of Glaucophyta. Similarly, relatively ancient counterparts such as LRR-RLPs, WAK-RLPs, LysM-RLPs, Malectin-RLPs, and

L-lectin-RLPs are found in both Glaucophyta and Rhodophyta. In contrast, RLKs emerged more recently. Green algae harbour WAK-RLKs, Malectin-RLKs, and G-lectin-RLKs, and LysM-RLKs, L-lectin-RLKs, and Duf-26-RLKs are exclusive to Embryophytes (Fig. 2). Except for LRR-RLPs, all six families of RLP are basal to the RLK families. This suggests the intriguing possibility that some RLKs may have evolved directly from RLPs through the integration of kinase domains. To test this hypothesis, we aligned the G-lectin-, WAK-, Duf26-, L-lectin-, LysM- and Malectin-ectodomains from the ectodomain-only proteins, RLPs, and RLKs within either a subset of 25 species or all 350 species (Supplementary Fig. 4). In some cases, we observed proportions of RLKs (such as WAK-RLKs) that likely have directly evolved from the RLPs (Supplementary Fig. 4a). In other cases (such as the LysM-RLKs; Supplementary Fig. 4c), it is not clear whether the RLKs evolved from an ectodomain-only protein or an RLP. Thus, we concluded that RLKs have emerged from either the RLPs or directly from ectodomain-only proteins. Within the global ectodomain sequence similarity trees, we also observed RLKs (the LysM-RLKs and Malectin-RLKs; Supplementary Fig. 4d, f) that have emerged from two independent events. While other RLKs (WAK-, G-lectin-, L-lectin- and Duf26-RLKs; Supplementary Fig. 4b, h, j, l) likely to have emerged from RLPs or ectodomain-only proteins through a series of domain-swapping events. Such events likely occurred at various stages of plant evolution, given the presence of RLKs in diverse plant lineages[14].

We also examined the expansion patterns of different receptor classes across various plant lineages. Our analysis involved calculating the median percentage (%) of cell-surface receptors families in (i) Glaucophyta and Rhodophyta, (ii) green algae, and (iii) Bryophytes to determine the percentage increase (% increase; see methods and Supplementary Note 2) from Glaucophyta and Rhodophyta to green algae; from green algae to Embryophytes; and between Bryophytes and Tracheophytes. We observed substantial expansions in specific receptor families across these lineages. Green algae exhibited a significant expansion of LRR-RLKs, while Embryophytes displayed expansions in LRR-RLPs, LRR-RLKs, WAK-RLKs, and G-lectin-RLKs. Tracheophytes had further expansions in LRR-RLPs, WAK-RLKs, Malectin-RLKs, G-lectin-RLKs, and Duf26-RLPs (Fig. 2; Supplementary Fig. 3). Overall, RLKs demonstrate greater expansion compared to RLPs, with notable expansions observed in LRR-RLK, WAK-RLK, and G-lectin-RLK. In addition, the LRR-RLP family has also significantly expanded throughout the plant lineage. These findings align with the substantial size of these receptor families and their involvement in recognising pathogens (Fig. 1i; Supplementary Fig. 1b). LRR-RLKs are

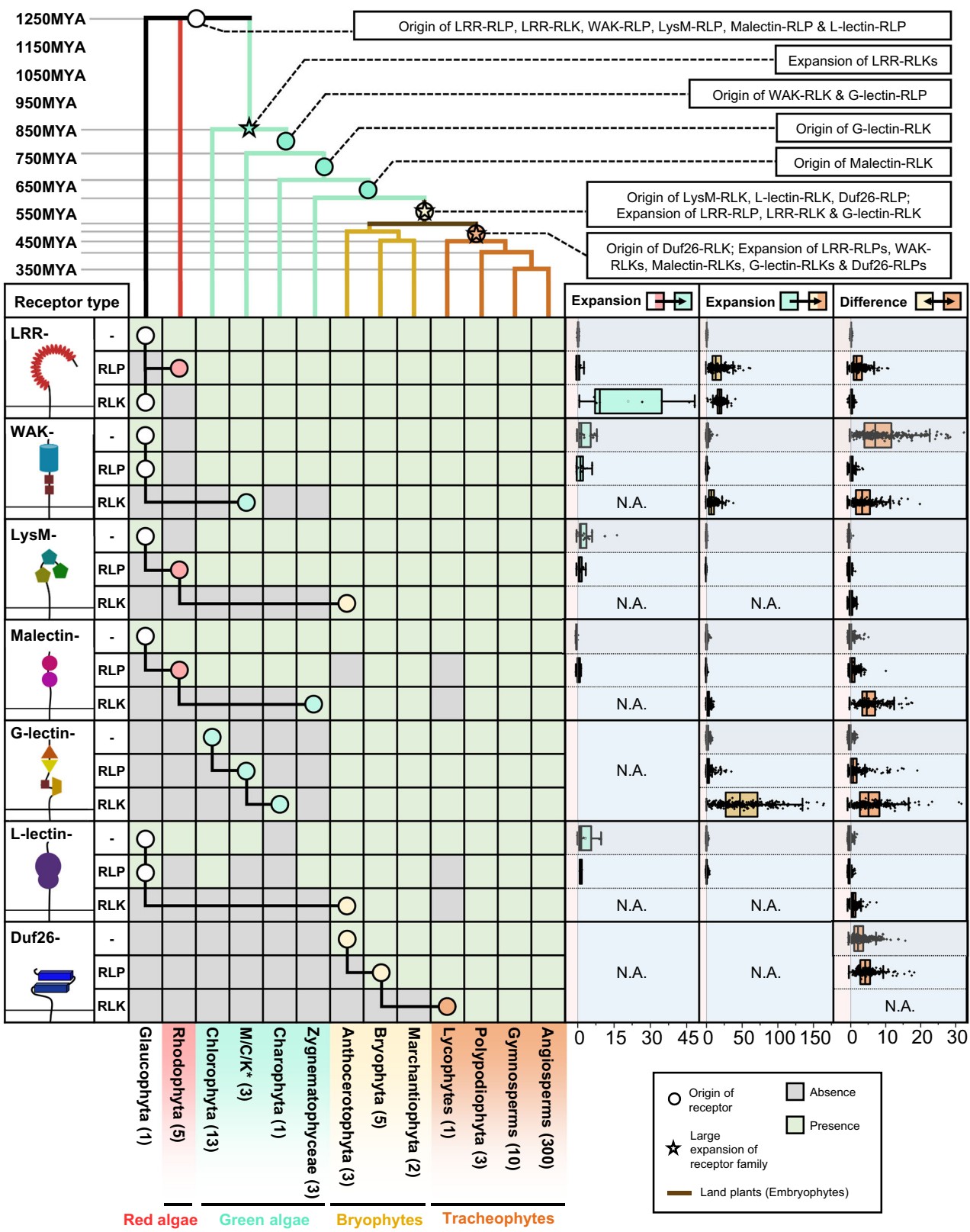

classified into 20 subgroups based on their kinase domains, with subgroup XII specifically implicated in PAMP recognition[15]. In particular, the LRR-RLK-XII subgroup exhibits a considerably higher expansion rate compared to other subgroups (Supplementary Fig. 5), reinforcing the idea that cell surface immune receptors underwent extensive expansions as the plant lineage diversified and evolved to adapt to a wide range of environments.

## The origin and expansion of PTI-signalling components in the plant lineage

Upon ligand or elicitor perception, PRRs undergo dimerisation or form heteromeric complexes with other LRR-RLK co-receptors, such as BAK1 (a member of the Somatic embryogenesis receptor-like kinases (SERK) family) and SOBIR1. This spatial arrangement brings the cytoplasmic kinase domains (from co-receptors and/or receptors) in close

**Fig. 2 | The origin and expansion of cell-surface receptors in plants.** The top panel represents a sequence similarity tree of multiple algal and plant lineages. Circles (○) and stars (☆) indicate the origin and expansion of receptor families. The timescale (in millions of years; MYA) of the sequence similarity tree was estimated by TIMETREE5[99]. The bottom panel represents the presence or absence of different receptor classes in algal and plant lineages. '-' represents ectodomains with no transmembrane or kinase domain, 'RLP' represents ectodomains with a transmembrane domain but no kinase domain, 'RLK' represents ectodomains with both transmembrane and kinase domains. *M/C/K represents Mesostigmatophyceae, Chlorokybophyceae, and Klebsormidiophyceae. The number of species available from each algal and plant lineage is indicated by the numbers within respective boxes. A grey box indicates the absence of receptors and a green box indicates their presence in each lineage. The origin of a receptor is indicated with a circle (○). The origins of '-', 'RLP', and 'RLK' are connected by black lines. Expansion rates of receptor classes are indicated by boxplots. The percentages (%) of cell-surface receptors from each genome were calculated as (number of identified genes/ number of searched genes × 100). Next, the percentages from each species within a lineage (e.g., Rhodophtya or green algae) were grouped and the median percentage was calculated. Median value was used instead of mean to avoid outliers within the lineages. The expansion rate within a species is calculated by ((% cell surface receptors in that species)-(median))/(median). The cyan boxplot represents the expansion rate from Glaucophyta and Rhodophyta to green algae (LRR, $n = 20$;

LRR-RLP, $n = 13$; LRR-RLK, $n = 9$; WAK, $n = 16$; WAK-RLP, $n = 19$; WAK-RLK, n = 0; LysM, $n = 20$; LysM-RLP, $n = 16$; LysM-RLK, $n = 0$; Malectin, $n = 9$; Malectin-RLP, $n = 6$; Malectin-RLK, $n = 0$; G-lectin, $n = 0$; G-lectin-RLP, $n = 0$; G-lectin-RLK, $n = 0$; L-lectin, $n = 8$; L-lectin-RLP, $n = 2$; L-lectin-RLK, $n = 0$; Duf26, $n = 0$; Duf26-RLP, $n = 0$; Duf26-RLK, $n = 0$). The yellow boxplot represents the expansion rate from green algae to Embryophytes (LRR, $n = 324$; LRR-RLP, $n = 324$; LRR-RLK, $n = 324$; WAK, $n = 316$; WAK-RLP, $n = 324$; WAK-RLK, n = 323; LysM, $n = 323$; LysM-RLP, n = 314; LysM-RLK, $n = 0$; Malectin, $n = 321$; Malectin-RLP, $n = 294$; Malectin-RLK, $n = 319$; G-lectin, $n = 319$; G-lectin-RLP, $n = 315$; G-lectin-RLK, $n = 322$; L-lectin, n = 277; L-lectin-RLP, $n = 314$; L-lectin-RLK, $n = 0$; Duf26, $n = 0$; Duf26-RLP, $n = 0$; Duf26-RLK, $n = 0$) and the orange boxplot represents the differences between early land plants to Tracheophytes (LRR, $n = 314$; LRR-RLP, n = 314; LRR-RLK, $n = 314$; WAK, $n = 311$; WAK-RLP, $n = 314$; WAK-RLK, $n = 314$; LysM, $n = 313$; LysM-RLP, $n = 304$; LysM-RLK, $n = 312$; Malectin, $n = 311$; Malectin-RLP, n = 292; Malectin-RLK, $n = 311$; G-lectin, $n = 309$; G-lectin-RLP, $n = 307$; G-lectin-RLK, $n = 313$; L-lectin, $n = 267$; L-lectin-RLP, $n = 306$; L-lectin-RLK, $n = 312$; Duf26, $n = 312$; Duf26-RLP, $n = 313$; Duf26-RLK, $n = 0$). Light blue area represents expansion and light pink area represents contraction of the gene family. X-axis values represent expansion rate (×). Values larger than 0 indicate expansion; values equal to 0 indicate no expansion, and values below 0 indicate contraction. Boxplot elements: centre line, median; bounds of box, 25th and 75th percentiles; whiskers, 1.5 × IQR from 25th and 75th percentiles. For details, refer to the methods.

proximity, initiating a cascade of auto- and trans-phosphorylation events[16]. The activated receptor complex subsequently phosphorylates members of the cytoplasmic receptor-like kinases subgroup VII (RLCK-VII)[17], which, in turn, phosphorylate various cytoplasmic kinases, such as the mitogen-activated protein kinase kinase kinases (MAPKKKs), calcium-dependent protein kinases (CDPKs) and plasma membrane-associated proteins, such as cyclic nucleotide-gated channels (CNGCs), hyperosmolality-gated calcium-permeable channels (OSCAs), and NADPH oxidases (RBOHs)[16]. The phosphorylation of these proteins collectively triggers transcriptional reprogramming and physiological changes, such as cytoplasmic calcium influx and the accumulation of reactive oxygen species (ROS)[18]. These physiological responses effectively hinder pathogen proliferation during infection (Fig. 2a; Supplementary Figs. 6 and 7).

We identified cell-surface co-receptors and signalling components from the 350 genomes and determined their absence or presence across the plant lineage (Supplementary Fig. 7). SERKs, acting as cell-surface co-receptors for multiple LRR-RLKs and LRR-RLPs are present in Zygnematophyceae and Embryophytes[19] (Fig. 3b; Supplementary Fig. 8a, b), suggesting their emergence during or prior to the appearance of land plants. Immune-related LRR-RLPs lack intracellular kinase domains, thus require another LRR-RLK co-receptor, SOBIR1, to activate downstream signalling[20]. Similar to BAK1, SOBIR1 is also present in Embryophytes (Fig. 3b). Thus, co-receptors for cell-surface receptors likely evolved during or before the emergence of land plants. On the other hand, cytoplasmic kinases (RLCKs, CDPKs, MAPKKKs, MAPKKs, and MAPKs) are ancient, as are the PM-localised downstream signalling components (CNGCs, OCSAs, and RBOHs), found across all plant lineages (Fig. 3b; Supplementary Fig. 8). Although the exact function of these proteins in algal species remains unclear, their immune-related orthologs are present in green algae (Fig. 3b; Supplementary Fig. 8. This suggests that they underwent specialisation within the immune activation pathway prior to the emergence of land plants. The EP proteins (EDS1, PAD4, and SAG101) and RPW8-NLRs (NRG1 and ADR1) that are essential for both TIR-NLR and LRR-RLP mediated-immunity[21,22], are only present in gymnosperms and angiosperms (seed plants)[23]. Considering the ancient nature of the LRR-RLPs, it is plausible that EP-protein and helper-NLRs were integrated into the LRR-RLP-signalling pathway, forming a robust immune network in seed plants.

Our investigation of the expansion rate of signalling components within the plant lineage indicated an expansion of CDPKs in green algae and expansion of RLCK-VIIs in Tracheophytes (Fig. 3b; Supplementary Fig. 8; Supplementary Note 2). However, other families of signalling components exhibit more limited expansions, compared to cell-surface receptors. This is also consistent with the considerably larger family sizes of cell-surface receptors and NLRs in comparison to the signalling components (Supplementary Fig. 9). Furthermore, we examined the correlation between the percentages of signalling components and PRRs (LRR-RLK-XIIs + LRR-RLPs) across genomes. Except for CNGCs, EP proteins, and RPW8-NLRs (0.6 > Pearson's $r > 0.3$), most signalling component families do not exhibit co-expansion or co-contraction with PRRs (Supplementary Fig. 9). Thus, we concluded that plants are more likely to evolve new receptors rather than downstream signalling components for adaptation. The RLCK-VIIs are further classified into ten subgroups which are differentially required for RLKs and RLPs to activate downstream responses[17,24–27] (Supplementary Figs. 6 and 7). Similarly, CDPKs fall into 4 subgroups (Supplementary Fig. 8). RLCK-VII and CDPK subgroup members are differentially required by different PRRs to activate downstream responses[28–31]. Pathogens often target RLCK-VIIs through secreted effectors to suppress immunity[32–34]. Thus, redundancy among RLCK-VII subgroups serves as a protective mechanism for the downstream signalling pathway against effector targeting. In addition, plants have evolved RLCK-VII pseudokinases, or 'decoys', to guard functional RLCKs through NLRs[34–38]. Together, it has become apparent that the expansion of RLCK-VII families may have been driven by pathogenic pressure, thereby contributing to the enhanced robustness of the immune signalling network.

## Immunity- and development-related cell surface receptors share a common origin

We sought to understand how cell-surface receptors evolved to be specialised in immunity. To achieve this, we decided to trace the evolutionary origin of LRR-RLPs in plants. Among RLPs, LRR-RLPs constitute the largest family, comprising more than twenty characterised members that perceive PAMPs or apoplastic effectors to activate immunity (Supplementary Fig. 10). The ectodomain of LRR-RLPs encompasses additional domains known as N-loop outs (NLs) and island domains (IDs) interspersed between the LRR motifs[39,40]. Typically, NLs are located closer to the N-termini of the ectodomain, whereas IDs are positioned closer to C-termini[39,40]. NLs are present in most immunity-related LRR-RLPs while IDs are present in all immunity-related LRR-RLPs (Supplementary Figs. 10 and 11). NL positioning is

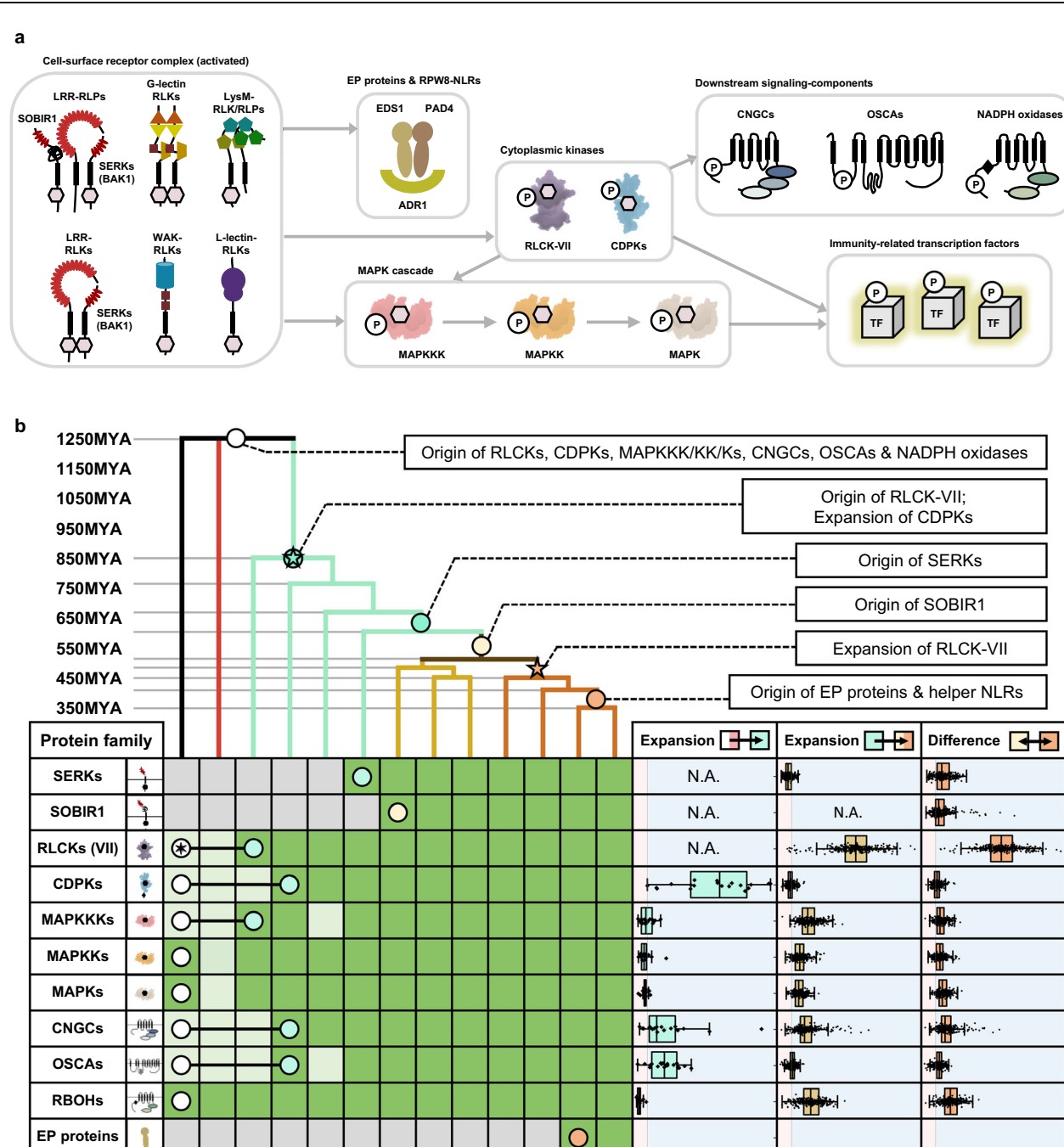

relatively more flexible, occurring either before the first LRR motif or between the first few LRR motifs. Conversely, ID positioning is less flexible, located mostly before the last 4 LRR motifs within the ecto-domain (Supplementary Figs. 10 and 11). This observation implies a functional necessity for the specific placement of IDs in LRR-RLPs.

To investigate the functional necessity of IDs, we analysed ecto-domains of LRR-RLPs and LRR-RLKs from 350 species (113,794) (Fig. 4a

and Supplementary Fig. 12a, b). Employing multiple prediction pro-grams, we identified gaps between LRR motifs ranging from 10–29 or 30–90 amino acids (AA) (Supplementary Fig. 12a). Since NLs typically span 6–30AA and IDs range from around 40–75AA, we focused on small gaps (10–29AA) corresponding to NLs, and large gaps (30–90AA) indi-cative of IDs (Supplementary Fig. 10). Small or large gaps are relatively infrequent in LRR-RLKs (10.6% and 5.43%, respectively) (Fig. 4a).

**Fig. 3 | The origin and evolution of cell-surface receptor signalling component in plants. a** Schematic figure represents the simplified PTI signalling pathway in plants. Coloured hexagons on RLKs indicate activated kinases. For details, refer to Supplementary Fig. 6. **b** The top panel is a sequence similarity tree of multiple algal and plant lineages. Circles (○) and stars (☆) indicate the origins and expansion of receptor families, respectively. The timescale (in million years; MYA) of the sequence similarity tree was estimated by TIMETREE5[99]. The bottom panel displays the presence or absence of receptor classes in different algal and plant lineages. *M/C/K represents Mesostigmatophyceae, Chlorokybophyceae, and Klebsormidiophyceae. The number of available species from each algal and plant lineage is indicated within the respective boxes. A grey box indicates the absence, while a green box indicates the presence of a given protein family in each lineage. Dark green indicates the presence of orthologs of immunity-related (PTI) signalling components within that protein family (see also Supplementary Fig. 8). The origin of a protein family is indicated with a circle (○), followed by another circle indicating the origin of the orthologs of PTI-signalling component. Expansion rates of PTI-signalling component families are indicated by boxplots. The percentages (%) of signalling components from each genome were calculated as (number of identified genes/number of searched genes × 100). Next, the percentages from each species within a lineage (e.g, Rhodophtya or green algae) were grouped and the

median percentage was calculated. Median value was used instead of mean to avoid outliers within the lineages. The expansion rate within a species is calculated by ((% signalling components in that species)-(median))/((median)). The cyan boxplot represents the expansion rate from Glaucophyta and Rhodophyta to green algae (SERKs, $n = 0$; SOBIR1, $n = 0$; RLKCs (VII), $n = 0$; CDPKs, $n = 20$; MAPKKKs, $n = 20$; MAPKKs, $n = 20$; MAPKs, $n = 20$; CNGCs, $n = 19$; OSCAs, $n = 19$; RBOHs, $n = 10$; EP proteins, $n = 0$; RPW8-NLRs, $n = 0$). The yellow boxplot represents the expansion rate from green algae to Embryophytes (SERKs, $n = 316$; SOBIR1, $n = 0$; RLKCs (VII), $n = 324$; CDPKs, $n = 324$; MAPKKKs, $n = 324$; MAPKKs, $n = 324$; MAPKs, $n = 324$; CNGCs, $n = 324$; OSCAs, $n = 322$; RBOHs, $n = 324$; EP proteins, $n = 0$; RPW8-NLRs, $n = 0$) and the orange boxplot represents the differences between early land plants to Tracheophytes (SERKs, $n = 307$; SOBIR1, $n = 309$; RLKCs (VII), $n = 314$; CDPKs, $n = 314$; MAPKKKs, $n = 314$; MAPKKs, $n = 314$; MAPKs, $n = 314$; CNGCs, $n = 314$; OSCAs, $n = 312$; RBOHs, $n = 314$; EP proteins, $n = 0$; RPW8-NLRs, $n = 0$). Light blue area represents expansion and light pink area represents contraction of the gene family. X-axis values represent expansion rate (×). Values larger than 0 indicate expansion; values equal to 0 indicate no expansion, and values below 0 indicate contraction. Boxplot elements: centre line, median; bounds of box, 25th and 75th percentiles; whiskers, 1.5 × IQR from 25th and 75th percentiles. For details, refer to the methods.

---

In contrast, both small and large gaps are more prevalent in LRR-RLPs (28.3% and 61.6%, respectively). Furthermore, both LRR-RLKs and LRR-RLPs typically have only one gap, which can be either small or large (Supplementary Fig. 12c, d). Our analysis also showed that small gap positions within the ectodomains of both LRR-RLKs and LRR-RLPs are not fixed, but may be distributed randomly. Conversely, larger gaps are predominantly positioned before the last four LRR motifs in the ectodomain (51.2% for LRR-RLKs and 86.9% for LRR-RLPs) (Fig. 4b, c and Supplementary Fig. 13). Thus, our findings suggest functional requirement for IDs to be positioned before the last four LRRs.

Analysis of the distribution of LRR-RLK subgroups with IDs indicated that over 55% belong to the Xb subgroup (Fig. 4d). Furthermore, 94.7% of LRR-RLKs with IDs positioned before the last four LRRs (ID + 4LRR) belong to the Xb subgroup (Fig. 4d), suggesting that both LRR-RLK-Xb and LRR-RLPs share the ID + 4LRR motif. Among the characterised members of LRR-RLK-Xb are important components of growth and development regulation, including the BRASSINOSTEROID INSENSITIVE 1 (BRI1) family members, PHYTOSULFOKIN RECEPTOR 1 (PSKR1) and PSY1 RECEPTOR (PSY1R) family members, EXCESS MICROSPOROCYTES1 (EMS1) and NEMATODE-INDUCED LRR-RLK 1 (NILR1/GRACE)[41–45]. Interestingly, we observed that both LRR-RLPs and LRR-RLK-Xbs with ID + 4LRR motifs are present in land plants (Embryophyte) but not in other lineages (Fig. 4e). Considering the similarity in structural motifs between LRR-RLPs and LRR-RLK-Xbs, it is likely that these two receptor classes share a common origin.

To test this hypothesis, we conducted phylogenetic and structural analyses on the ectodomains of LRR-RLPs and LRR-RLKs. First, we aligned the last four LRR motifs (referred to as the C3 region[46]) from LRR-RLKs and LRR-RLPs (60,240). LRR-RLPs and LRR-RLK-Xbs with an ID + 4LRR cluster together in the C3 sequence similarity tree, indicating a common origin (Fig. 4f). Within this cluster (indicated as the ID + 4LRR clade), we observed distinct subclades, including the BRI1/BRI1-LIKE (BRL) clade, the PSKR/PSY1R clade, and mostly LRR-RLPs (Fig. 4g). The closely similar C3 regions indicate a conserved function. BRI1 recognises brassinosteroids (BRs) and interacts with the co-receptor BAK1 to induce BR responses[47,48]. Similarly, PSKR1 perceives phytosulfokine (PSK) and interacts with the co-receptor SERK1/2. LRR-RLPs, on the other hand, perceive PAMPs or apoplastic effectors and engage both co-receptor BAK1 and SOBIR1 to initiate immune responses[49]. Structural studies have elucidated the interaction mechanism of BRI1, PSKR1, and the LRR-RLP RXEG1 with SERKs[48,50–54] (Fig. 5a–c, Supplementary Fig. 14). Interestingly, the C3 regions of BRI1, PSKR1, and RXEG1 play a crucial role in SERK interactions[48,50,52,53], differentiating them from other LRR-RLKs. BRI1, PSKR1 and RXEG1

contain specific amino acid residues within the terminal two LRR motifs (within the C3 region) to interact with BAK1/SERK1 (Fig. 5a–c), while the flagellin peptide (flg22) receptor FLS2 (an LRR-RLK-XII), relies on the 3$^{rd}$–12$^{th}$ last LRR for BAK1 (SERK3) interactions[55] (Supplementary Fig. 14a). The N-terminal region (or N-terminal cap) of BAK1 or SERK1 is important for FLS2, BRI1, PSKR1, and RXEG1 interactions, while the 1$^{st}$ LRR inner surface of BAK1 or SERK1 is primarily involved in BRI1, PSKR1, and RXEG1 interactions, and the 2$^{nd}$ and the 4$^{th}$ LRR inner surface of BAK1 is involved in FLS2 interactions (Supplementary Fig. 14b). There are also striking similarities in interaction network maps between PSKR1-SERK1 and RXEG1-BAK1 interactions (Fig. 5a, b). Additionally, the C3 region of BRI1 participates in both BR binding[50,52] and co-receptor binding, whereas the C3 region of PSKR1 and RXEG1 exclusively engages in co-receptor interactions (SERK1/2 and BAK1, respectively)[48,53]. By aligning the C3 regions of various LRR-RLKs, including BRI1/BRL-orthologs, PSKR orthologs, PSY1R, and multiple LRR-RLPs, overlapping residues that are required for SERK interactions within PSKR/PSY1R, and LRR-RLP clades can be discerned (See Results section 'Specialisation of cell-surface receptors in different biological processes' for alignment). For example, the glutamic acid e residue at the second position of the penultimate LRR motif is involved in both PSKR1-SERK1 and RXEG1-BAK1 interactions[53]. This E residue is highly conserved in both clades (PSKR/PSY1R clade: 69.9%, LRR-RLP: 86%). Similarly, the phenylalanine (F) residue at the last position of the penultimate LRR motif contributes to both PSKR1-SERK1 and RXEG1-BAK1 interactions[48]. This F residue is conserved in the BRI1/BRL (62.2%), PSKR/PSY1R (99.7%), and LRR-RLP (64.1%) clades. There is also a conserved motif crucial for SERKs interactions within the last LRR motifs of BRI/BRL, PSKR/PSY1R, and LRR-RLP clades. This motif, Glutamine-x-x-Threonine/Serine (QxxT/S) loop, is conserved in BRI1/BRL (Q:86.6%; T/S:98.9%), PSKR/PSY1R (Q:99.9%; T/S:92.5%) and LRR-RLP (Q:91.8%; T/S:88.4%) clades, but it is not conserved in other LRR-RLPs outside of the ID + 4LRR clade or other LRR-RLK subgroups. Structural analysis of BRI1-BAK1, PSKR1-SERK1/2, and RXEG1-BAK1 further supports the importance of residues within and around the QxxT/S loop for SERK interactions[48,50,52,53] (Fig. 5a–c). In conclusion, our findings indicate that the C3 regions of LRR-RLK-Xb and LRR-RLPs share a conserved function for interacting with SERKs.

To further validate the functional conservation of C3 regions in LRR-RLK-Xb and LRR-RLPs, we performed functional analysis of the QxxT/S motif in the LRR-RLP RLP23 from *A. thaliana*. RLP23 forms heteromeric complexes with the LRR-RLK co-receptor SOBIR1, and upon the perception of the nlp20 peptide, BAK1 is recruited into the complex, leading to activation of the SOBIR1 KD to induce

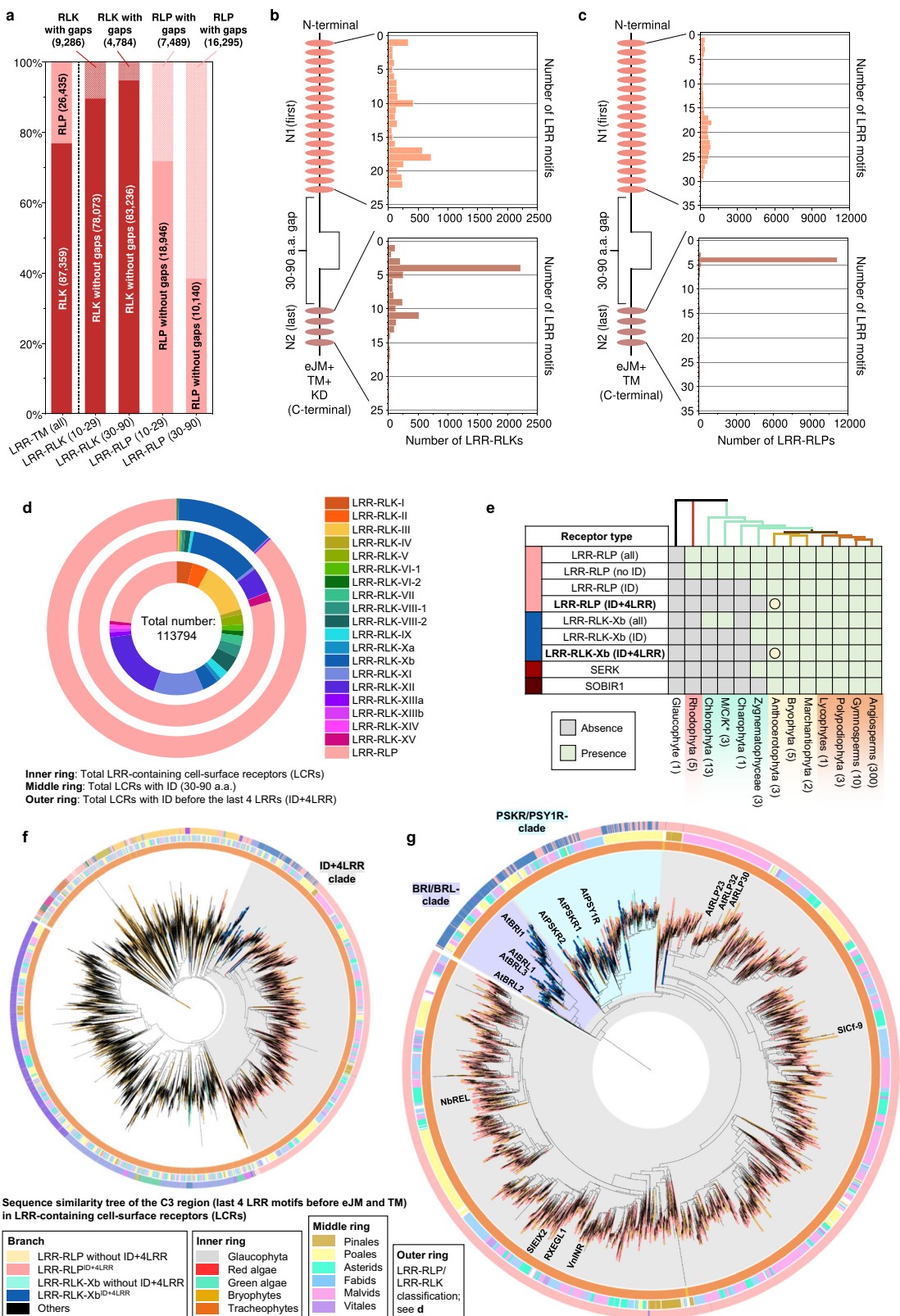

**Inner ring**: Total LRR-containing cell-surface receptors (LCRs)
**Middle ring**: Total LCRs with ID (30-90 a.a.)
**Outer ring**: Total LCRs with ID before the last 4 LRRs (ID+4LRR)

Sequence similarity tree of the C3 region (last 4 LRR motifs before eJM and TM) in LRR-containing cell-surface receptors (LCRs)

immunity[20,49,56]. Similar to RXEG1 (with TQLQT), RLP23 possesses a TQITG motif (TQxxx), while LRR-RLK-Xb members, such as PSY1R, PSKR1, PSKR2, and BRI1, feature TQFDT, GQFQT, GQFYS, and GQFET motifs, respectively (all QxxT/S). Notably, the LRR-RLK-XII member EFR lacks the QxxT/S motif in that position, having GVFRN instead (Fig. 5d, e). We generated chimeric constructs of RLP23 with the terminal LRR motifs swapped between PSY1R, PSKR2, BRI1, and EFR

(Fig. 5e, f). By immuno-precipitation assays, we tested the ability of these chimeras to interact with BAK1 upon ligand perception. Both wildtype RLP23 (WT) and RLP23-PSY1R (PY) chimeras can interact with BAK1 upon nlp20 treatment, whereas RLP23-BRI1 (BR) and RLP23-EFR (EFR) cannot (Fig. 5g). This suggests that the terminal LRR motif of RLP23 and PSY1R bind BAK1 in a similar manner. All chimeras can interact with SOBIR1 regardless of the presence of nlp20, indicating

**Fig. 4 | The origin and evolution of LRR$^{ID+4LRR}$ in plants. a** Distribution of LRR-RLKs and LRR-RLPs in LRR-containing cell surface receptors (LRR-TM) from 350 plant species, and the distribution of LRR-RLPs and LRR-RLKs with or without gaps of 10-29 amino acids (10-29) or 30-90 (30-90) amino acids. **b, c** Position of large gaps (IDs; 30–90 AA) in **b** LRR-RLKs and **c** LRR-RLPs with a single large ID. N1 represents the number of LRR motifs before the IDs and N2 represents the number of LRR motifs after the ID. For positions of gaps in LRR-RLKs and LRR-RLPs with multiple gaps, refer to Supplementary Fig. 13. **d** The concentric ring pie chart presents the percentage of LRR-containing cell-surface receptors (PRRs) from 350 species. The inner ring represents all LRR-containing cell-surface receptors (113,794); the middle ring represents LRR-containing PRRs with ID (20,556); the outer ring represents LRR-containing PRRs with an ID preceding the last 4 LRR (ID + 4LRR) at the C terminus (16,885). **e** The presence or absence of receptor classes in various algal and plant lineages. *M/C/K represents

Mesostigmatophyceae, Chlorokybophyceae, and Klebsormidiophyceae. A grey box indicates the absence, and a green box indicates the presence of a given receptor class in each lineage. The origin of LRR-RLP and LRR-RLK-Xb with ID + 4LRR is indicated with a circle (○). **f** Sequence similarity tree of the C3 region (last 4 LRRs) from all LRR-containing cell-surface receptors of 350 species. Branches are colour-labelled as indicated. The inner ring and middle ring indicate the lineage and subclass/order of the corresponding protein (species) from the branch. Outer ring represents the LRR-RLP or LRR-RLK classification, which is indicated in **d**. The light grey area indicates clustering of LRR-RLK-Xb and LRR-RLP with ID + 4LRR. The pruned sequence similarity tree on the right **g** corresponds to the light grey area in the left tree, with clades labelled in dark grey areas accordingly. Characterized LRR-RLK-Xb and LRR-RLP members are labelled. The BRI/BRL-clade and the PSKR/PSY1R-clades are also labelled.

that the last LRR motif may not be involved in SOBIR1 interactions (Fig. 5g). Furthermore, upon nlp20 treatment, both WT and PY can trigger immune responses, while K2 and BR cannot (Fig. 5h, i). We speculate that this may be due to the absence of a specific T residue before the Qxxx motif in PSKR2 and BRI1 (G instead of T), which is relatively less prevalent within the LRR-RLP clade (31.1%). Multiple *A. thaliana* LRR-RLPs contain this residue, but it is not in other studied LRR-RLPs, such as the tomato Cf proteins, suggesting that this residue evolved in some species after the divergence of LRR-RLK-Xb and LRR-RLPs. To test this hypothesis, we generated chimeric constructs of RLP23 with the terminal LRR motifs swapped to PSKR1 (K1), PSKR2 (K2), BRI1 (BR), and mutated the G residue into T (thereby K1$^T$, K2$^T$, BR$^T$; Fig. 5e, f). We tested the immune responses triggered by these chimeras following nlp20 treatment. Both K1, K1$^T$, and K2$^T$ induce relatively immune responses compared to WT and PY, while K2, BR, and BR$^T$ are unable to induce any immune responses (Fig. 5j, k). Both K1 and K1$^T$ weakly interact with BAK1 upon nlp20 treatment, while K2 does not interact with BAK1 (Fig. 5k). Thus, the residues around and within the TQxxx motif are both crucial for SERK interactions. We concluded that the C3 region in LRR-RLPs and some LRR-RLK-Xbs (such as PSY1R and PSKR1) interact with SERKs in a similar manner, while some LRR-RLK-Xbs (PSKR2 and BRI1) have evolved to interacts with SERKs in a slightly different manner. Nevertheless, our results strongly support the functional conservation of C3 regions in LRR-RLK-Xbs and LRR-RLPs, specifically their ability to interact with SERKs.

## On the origin of LRR-RLPs and LRR-RLK-Xbs

The functional conservation of C3 regions in LRR-RLK-Xbs and LRR-RLPs implies that the ectodomains of these two receptor families might share a common origin. To dissect the ectodomain origin of LRR-RLPs and LRR-RLK-Xbs, we investigated the relatedness of the IDs between these two groups of receptors. First, we aligned the IDs extracted from both LRR-RLKs and LRR-RLPs (20,246). Remarkably, the ID clusters of LRR-RLPs and LRR-RLK-Xb are found in close proximity, mirroring the C3 sequence similarity tree (Fig. 6a). Again, the BRI1/BRI1-LIKE (BRL) and PSKR/PSY1R clusters are in proximity to the LRR-RLPs (Fig. 6a, b). These results are also consistent with a previous report that PSKRs are closely related to some LRR-RLPs in Arabidopsis and rice[46]. To dissect the ectodomain origin of LRR-RLPs and LRR-RLK-Xbs. A recent review identified two conserved lysine k-containing motifs, Yx8KG and Kx5Y, in the ID of LRR-RLPs[40]. The lysine residue in the Kx5Y motif from RXEG1 is required for its interaction with BAK1[53]. We identified both Yx8KG and Kx5Y motifs in the extracted IDs (before the last 4LRR motifs) of LRR-RLKs and LRR-RLPs. More than 75% of IDs from LRR-RLPs contain at least one of these lysine motifs, while less than 5% of IDs from LRR-RLK-Xb have either motif (Fig. 6c). This is consistent with structural data indicating that IDs from BRI1 and PSKR1 employ distinct residues for their interactions with BAK1[48,50,52]. However, IDs from LRR-RLPs that are closely related to those from LRR-RLK-Xbs retain Kx5Y motifs (Fig. 5b). It is therefore possible that the

common ID ancestor may have originally harboured lysine motifs that were subsequently lost from the BRI1 and PSKR/PSY1R clades.

To further explore the relatedness of the IDs between LRR-RLK-Xb group and LRR-RLPs, we formed clusters of highly similar IDs and examined their subgroup/family affiliations. Overall, we identified more than 2,822 clusters, with the majority (2,734) consisting of LRR-receptors of a single subgroup/family (Supplementary Fig. 15). Among the 61 clusters containing LRR-receptors from two different sub-groups/families, 80% (49 clusters) consist of LRR-receptors from LRR-RLK-Xb and LRR-RLPs (Supplementary Fig. 15). The enrichment of LRR-RLK-Xb and LRR-RLP pairings provides further evidence for the relatedness of their ectodomains. The substantial number of LRR-RLP-only ID clusters (2,383) also suggests that the IDs of LRR-RLPs have undergone extensive evolution and diversification, in the process providing a broad scope for the recognition of PAMPs and apoplastic effectors. Most clusters contain relatively a small number of IDs, predominantly from species within the same order or family (Supplementary Data 2). Consequently, tracing back the origin of IDs is challenging due to their considerable diversity. We therefore propose that the IDs of LRR-RLK-Xbs and LRR-RLPs likely originated from a common ancestor, with the IDs of LRR-RLPs expanding and diversifying after the divergence of LRR-RLK-Xb and LRR-RLPs.

Within the BRI1/BRI1-LIKE (BRL) and PSKR/PSY1R clusters, we found LRR-RLP counterparts of BRI1, PSKR, and PSY1R. To further test their relatedness, we aligned the full ectodomains from LRR-RLKs and LRR-RLPs and extracted the clades containing BRI1, PSKR, and PSY1R from the sequence similarity tree (Supplementary Fig. 16a, b). Within the BRI1/BRL- and PSKR-ectodomain clades (Fig. 6e, h), we found ectodomains from LRR-RLPs that are highly similar to BRL1/BRL3 and PSKR2, with residues essential for BR and PSK binding, respectively[48,54] (Fig. 6d-l). Within the PSY1R-ectodomain subclade, we found multiple IDs from LRR-RLPs that share remarkable similarity with AtPSY1R (Supplementary Fig. 17b, c). AtRLP3 and AtPSY1R have over 70% sequence identity and 85% similarity in their ectodomains. Although PSY1R is not the receptor for PSY peptide[57], it is possible that RLP3, and PSY1R recognise similar or identical ligands[58]. Currently, the functions of these BRL-, PSKR- and PSY1R-like LRR-RLPs remain unclear. AtRLP3 confers resistance against the vascular wilt fungus *Fusarium oxysporum* f. sp. *matthioli*[58], while AtPSY1R is involved in growth and development[43]. We propose that these RLPs may either recognise endogenous molecules to activate growth and development, or participate in the recognition of pathogen-mimicking molecules to trigger immune signalling.

## Specialisation of cell-surface receptors in different biological processes

Given the common origin of the ectodomains of LRR-RLPs and LRR-RLK-Xbs, these receptors must have undergone specialisation in immune- and developmental processes following their divergence. While LRR-RLK-Xbs activate downstream responses through the Xb kinase domain,

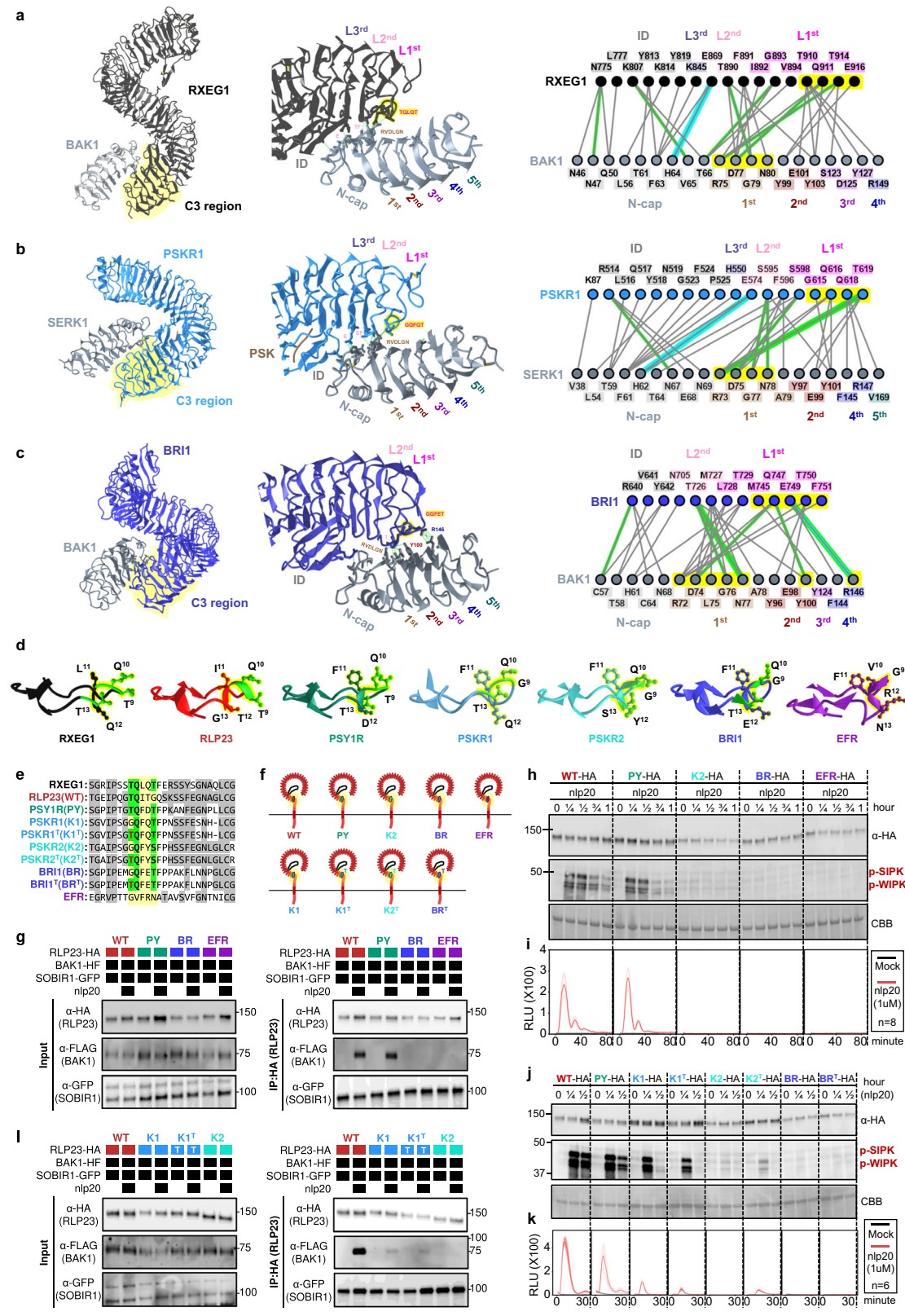

LRR-RLPs recruit SOBIR1 to trigger immunity[49,56]. Because LRR-RLPs lack a kinase domain, the juxtamembrane (JM) and TM regions may activate immune responses. We therefore aligned the C3, eJM (JM region before TM), TM, and cJM (JM region after TM, but with the absence of kinase domain from RLKs) regions from LRR-RLKs and LRR-RLPs (with a subset of 40, or the full set of 62,896 cell-surface receptors; Fig. 7a–c). Interestingly, the eJM-TM-cJM region of LRR-RLPs is not closely related to

LRR-RLK-Xbs (Fig. 7c). Previous studies have reported the requirement of Glycine-X-X-X-Glycine (GxxxG) motifs in SOBIR1 for its association with LRR-RLPs, and the potential contribution of negatively charged amino acids in the eJM region of LRR-RLPs to their interaction with SOBIR1[59]. Consistent with previous reports[59], LRR-RLPs are strongly negatively charged at the end of the eJM region, whereas LRR-RLKs, including SOBIR1, are positively charged at the end of the eJM region

**Fig. 5 | Functional characterization of the C3 region in LRR-RLPs and LRR-RLK-Xb. a–c** Structures and interaction interfaces of LRR-RLKs and LRR-RLPs with SERKs. Published structures of **a** NbRXEG1-NbBAK1[53], **b** AtPSKR1-AtSERK1[48], and **c** AtBRI1-AtBAK1[50] and are shown. The left panels show the full structure, and the middle panels show the interaction sites between LRR-RLKs or LRR-RLP and SERKs. Hydrogen bonds are indicated by green dotted lines, and salt bridges are shown as cyan dotted lines. The positions of LRR residues (counting from N to C for SERKs and counting from C to N for LRR-RLKs and LRR-RLP) are shown. Amino acid residues that are important for the interactions are labelled and the QxxT motifs are highlighted in yellow (red text). The right panel represents the 2D interaction network between SERKs and the receptors. Contacts/interactions are shown in grey lines, hydrogen bonds are shown in green lines, and salt bridges are shown in cyan lines. Amino acids are labelled in colours according to their positions in the LRR motifs (counting from N to C for SERKs and counting from C to N for LRR-RLKs and LRR-RLP l). Residues around and within the QxxT motifs in BRI1, PSKR1, and RXEG1 are highlighted in yellow. Residues in SERKs that are involved in the interactions with QxxT motifs are also highlighted in yellow. Structures were visualized in iCn3D[98]. For **a–c**, the interaction sites are calculated by iCn3D with the following thresholds: hydrogen bonds: 4.2 Å; salt bridges/ionic bonds: 6 Å; contacts/interactions: 4 Å. **d** Structure of the terminal LRR motif of *N. benthamiana* (Nb)RXEG1, *A. thaliana* (At)RLP23, AtPSY1R, AtPSKR1, AtPSKR2, AtBRI1 and AtEFR. Structures of NbRXEG1, AtPSKR1, and AtBRI1 were published[48,50–54]. Structures of AtRLP23, AtP-SY1R, AtPSKR2 and AtEFR were predicted by Alphafold2[97]. Ectodomains are visualised in iCn3D[98]. **e** Alignment of amino acids in the last LRR motifs from NbRXEG1, AtRLP23, AtPSY1R, AtPSKR1, AtPSKR1[T] (G > T), AtPSKR2, AtPSKR2[T] (G > T), AtBRI1, AtBRI1[T] (G > T), and AtEFR. Amino acid residues involved in the interaction between NbRXEG1 and BAK1 are highlighted in green. The QxxT motif positions are highlighted in yellow. Amino acids with similar properties to AtRLP23 are highlighted in grey. **f** Design of AtRLP23 chimeras. The last LRR motif of AtRLP23 is exchanged with either AtPSY1R, AtPSKR1, AtPSKR2, AtBRI1, or AtEFR. The glycine g residues in AtPSKR1, AtPSKR2, AtBRI1 have also been mutated to threonine (T). **g, l** Immuno-precipitation to test interactions between AtRLP23 chimeras, AtBAK1 and AtSOBIR1. Nb leaves expressing the indicated constructs were treated with either mock or 1 µM nlp20 for 5 min. **h–k** Functionality testing of AtRLP23 chimeras. Nb leaves expressing the indicated constructs were treated with 1 µM nlp20 and samples were collected at indicated time points. Phosphorylation of NbSIPK and NbWIPK was detected with p-P42/44 antibody. **i, k** Nb leaf discs expressing the indicated constructs were collected and treated with either mock or 1 µM nlp20, and ROS production was measured for indicated time points. For **i** and **k**, solid line, mean; shaded band, s.e.m. RLU, relative light units. For details of experiential design in **g–l**, refer to the methods section. For **g, h, j** and **l**, the experiments were repeated at least twice with similar results.

(Fig. 7a). Most LRR-RLPs have a single GxxxG motif, with some having two (GxxxGxxxG) or three (GxxxGxxxGxxxG) consecutive motifs (Fig. 7a). Conversely, GxxxG motifs are relatively less common in LRR-RLKs, but can be found in SOBIR1, BAK1, and the PSKR/PSY1R clade (Fig. 7a). We further examined overall charges in the eJM region of the LRR-RLP and LRR-RLK families. Most LRR-RLK subgroups feature positively charged eJM regions, whereas LRR-RLPs have negatively charged eJM regions. Importantly, negatively charged eJM regions are only present in LRR-RLPs within the ID + 4LRR clade (Fig. 7d). More than 80% of LRR-RLPs within the ID + 4LRR clade contain GxxxG motifs in their TMs. Moreover, GxxxGxxxG motifs are primarily found in LRR-RLPs and are relatively rare in LRR-RLKs (Fig. 4d). Interestingly, both GxxxG and GxxxGxxxG motifs are relatively enriched in the PSKR/PSY1R clade, indicating that the TM regions of LRR-RLPs and PSYR/PSY1R clade might also share a common origin. (Fig. 7d).

To assess the functionality of eJM-TM-cytosolic regions in LRR-RLK-Xbs and LRR-RLPs, we generated multiple chimeras of eJM, TM, and cytosolic regions of BRI1 and RLP23 (Fig. 8a). In RLP23[c-BRI1], the cytosolic region (following TM) of BRI1 was swapped into RLP23. In RLP23[TM-BRI1], the TM+ cytosolic region of BRI1 was swapped into RLP23. In RLP23[eJM-BRI1], the eJM + TM + cytosolic region of BRI1 was swapped into RLP23 (Fig. 8a). Immuno-precipitation assays showed that both RLP23[WT] and RLP23[c-BRI1] exhibit constitutive interactions with SOBIR1, and specifically interact with BAK1 upon nlp20 treatment. RLP23[TM-BRI1] does not interact with SOBIR1 but still interacts with BAK1 upon nlp20 treatment, suggesting that the RLP23 TM region with GxxxG motifs is necessary for SOBIR1-, but not BAK1-interactions (Fig. 8b). Consistent with these immunoprecipitation assays, RLP23[WT] and RLP23[c-BRI1] can activate immune responses, while RLP23[c-BRI1] and RLP23[TM-BRI1] can activate developmental responses as indicated by the dephosphorylation of BRI1-EMS-SUPPRESSOR 1 (BES1)[60] (Fig. 8c; Supplementary Fig. 18). This confirms that the BRI1 kinase domain is specifically required for the activation of BR responses. Interestingly, RLP23[TM-BRI1] can also activate weak immune responses, likely independent of SOBIR1 (Fig. 8c, d). RLP23[eJM-BRI1] does not interact with SOBIR1 or BAK1, and is unable to activate either immune or developmental responses (Fig. 8c, d). These data suggest that the eJM region of RLP23 is necessary for RLP23-BAK1 interactions. This conclusion was reinforced by generating RLP23[eo-BRI1] chimera, in which the eJM region of RLP23 is replaced with the BRI1 eJM (Fig. 8a). RLP23[eo-BRI1] consistently accumulates less protein than RLP23[WT]. Moreover, RLP23[eo-BRI1] constitutively interacts with SOBIR1, but fails to interact with BAK1 with nlp20 treatment (Fig. 8e). RLP23[eo-BRI1] only weakly activates MAPKs and fails

to trigger ROS production with nlp20 treatment (Fig. 8f, g). Thus, we conclude that the eJM regions of LRR-RLPs are important for protein accumulation and interaction with BAK1 upon ligand perception. The eJM, TM, and cytosolic region of LRR-RLPs and LRR-RLK-Xbs have indeed specialized to recruit different proteins into the receptor complex, which allows them to activate differential downstream responses (Fig. 8h).

## Discussion

The ectodomains of LRR-RLPs and LRR-RLK-Xbs appear to share a common evolutionary origin, suggesting an ancestral adoption of the ID + 4LRR architecture for ligand recognition and interactions with co-receptors such as the SERKs. Since both LRR-RLPs and LRR-RLK-Xbs rely on the terminal 4LRR/C3 region for SERK interaction[48,50,52,53], it is plausible that the C3 region has undergone stabilising evolution to preserve this functionality (Fig. 8i). Consequently, a portion of the C3 region in LRR-RLK-Xb and LRR-RLP can be interchanged without loss of function. However, the remaining ectodomain, including the LRR motifs preceding the ID and the ID itself, has undergone diversification and adaptation to recognise distinct ligands (Fig. 8i). For example, the perception of a herbivore-associated peptide (inceptin) by inceptin receptor (INR) emerged specifically in legume species around 28 million years ago[61]. As a result of this rapid diversification, we were unable to trace back to the common ancestor of LRR-RLK-Xb and LRR-RLP. Notably, certain LRR-RLPs exhibit significant sequence similarities to LRR-RLK-Xbs (BRI1/BRL, PSKR, and PSY1R), though their specific functions remain to be explored. Following diversification, the eJM, TM, and cytosolic regions of LRR-RLK-Xb and LRR-RLP acquired distinct roles, primarily in development and immunity, respectively (Fig. 8i). The eJM and TM regions of LRR-RLP specifically facilitate constitutive interactions with SOBIR1 and ligand-dependent interactions with BAK1, whereas the LRR-RLK-Xb group lacks such specialisation. Since the recruitment of SOBIR1 to BRI1 would lead to immune activation, there should be negative selection for negatively charged eJMs and GxxxGs in the TM regions to prevent immune activation by LRR-RLK-Xbs. Given the different roles of various domains and regions of LRR-RLKs and LRR-RLPs, we propose a model in which different domains or regions of cell-surface receptors undergo modular evolution to either diversify or maintain their original functions. Modular evolution allows the specialisation of cell-surface receptors to recognise different ligands and to activate distinct downstream signal responses while maintaining interactions with co-receptors (Fig. 8i). This model is consistent with the observation that LRR-RLKs have undergone substantial structural evolution to generate novel receptors[14].

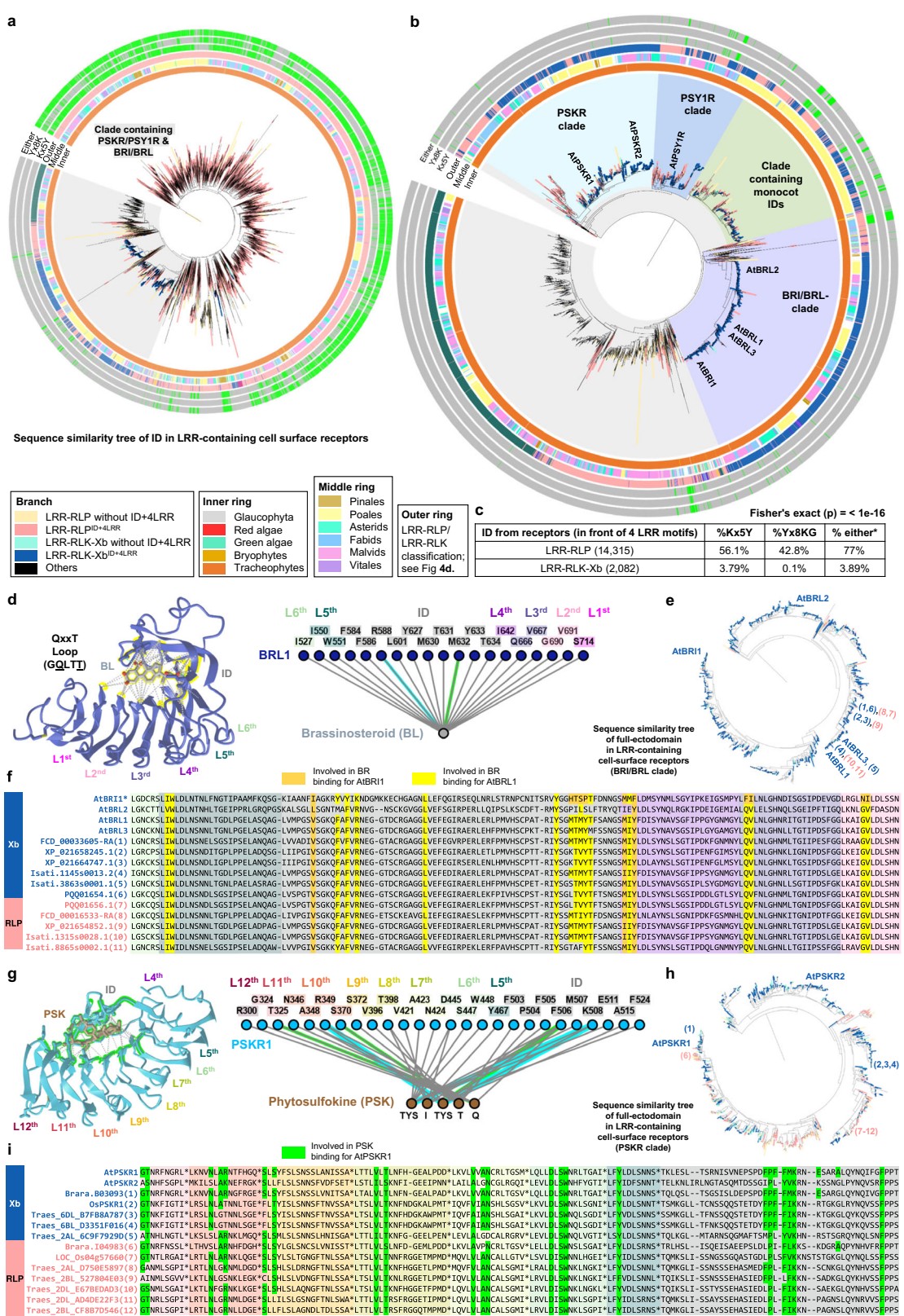

The presence of diverse cell-surface receptor classes and multiple downstream signalling components in algal species suggests that algae may have pathogen-sensing system (Fig. 5a), as indicated by the PAMPs triggering defence responses in some algal species[62] (Supplementary Note 2). Most cell-surface receptors and downstream signalling components in the Tracheophytes are conserved in Bryophytes. (Fig. 9a). Therefore, the most recent common ancestor of land plants is likely to

possess a considerable number of cell-surface receptors and the basic components of a signalling network. We also observed a significant expansion of cell-surface receptors families in land plants, which likely facilitated the adaptation to terrestrial environments (Fig. 9a). Our work has uncovered multiple evolutionary mechanisms underlying cell-surface receptors to facilitate plant adaptations: (i) Expansion of the number of receptors and their recognition specificity.

**Fig. 6 | On the origin of LRR-RLPs and LRR-RLK-Xbs. a** Sequence similarity tree of IDs extracted from all LRR-containing PRRs. Branches are labelled in colours as indicated. Grey clade represents clade that contains both PSKR/PSY1R & BRI/BRL family, this clade is shown in **b**. In **b**, characterized LRR-RLK-Xb and LRR-RLP members are labelled. The BRI/BRL-clade, the PSKR clade, the PSY1R clade, and a clade with monocot-only IDs are labelled in different colours. In **a**, **b**, inner ring (Inner), middle ring (Middle) and outer ring (Outer) are labelled as indicated. Outer ring represents LRR-RLP and LRR-RLKs classification shown in Fig. 4d. In the out-most three rings, the presence of Kx5Y (Kx5Y), Yx8K (Yx8K), or either Kx5Y or Yx8K (Either) in the IDs are indicated in green, and absence of these motifs is indicated in grey. **c** Percentage (%) of IDs (before the last four LRR motifs) from LRR-RLP and LRR-RLK-Xb with the Kx5Y, Yx8KG, or either Kx5Y or Yx8KG (*) motifs. The Fisher test (2-sided) was performed to compare the number/fraction of IDs with either Kx5Y or Yx8KG in LRR-RLPs against LRR-RLK-Xb. The calculated p-value stated here (<1e-16) is too low to be given exact number. Thus, the upper bound limit is stated instead. **d**, **g** Structures and interaction interfaces of AtBRI1-Brassinosteroid (BR) and AtPSKR1-Phytosulfokine (PSK). Published structures of **d** AtBRL1-BR[100], **g** AtPSKR1-PSK[48] are shown. The left panels show the interaction sites between the LRR-RLK-Xb receptors and their ligands. Contacts are indicated by grey lines, hydrogen bonds are indicated by green dotted lines, and salt bridges are shown as cyan dotted lines. The positions of LRR residues (counting from C to N) are shown. Amino acid residues that are important for the interactions are highlighted in yellow and green, respectively. The right panel represents the 2D interaction network between the LRR-RLK-Xb receptors and their ligands. Contacts/interactions are shown in grey lines, hydrogen bonds are shown in green lines, and salt bridges

are shown in cyan lines. Amino acids are labelled in colours according to their positions in the LRR motifs (counting from C to N). Structures were visualized in iCn3D[98]. **e**, **h** Sequence similarity trees of the full-ectodomains of **e** BRI1/BRL clade and **h** PSKR clade from 350 species. Branches are labelled in colours as indicated in **a**. These trees are extracted from the BRI1/BRL and PSKR branches from Supplementary Fig. 16. In **e**, **h**, characterized LRR-RLK-Xb members are labelled. The LRR-RLK-Xb and LRR-RLP members taken for the alignment in **f** and **i** are also labelled in blue numbers (LRR-RLK-Xb) and pink numbers (LRR-RLP), respectively. **f**, **i** Ectodomain and alignment of multiple LRR-RLK-Xb and LRR-RLP members within the BRI1/BRL-clade extracted from **e** and the PSKR-clade extracted from **h**. **f** The alignment of ectodomain from LRR-RLK-Xb (blue) and LRR-RLP (pink) members taken from the sequence similarity tree in **e**. The orange highlights indicate the amino acids residues required from BR binding in AtBRI1, and the yellow highlights indicate the amino acid residues required for BR binding in AtBRL1[100]. The yellow highlights are corresponding to the amino acids highlighted in the structure in **d**. The LRR motifs and ID in the alignment are indicated in colours shown in **d** (the interaction network; right panel). **i** The alignment of ectodomain from LRR-RLK-Xb (blue) and LRR-RLP (pink) members taken from the sequence similarity tree in **h**. The green highlights indicate the amino acids residues required from PSK binding in AtPSKR1[48]. The green highlights are corresponding to the amino acids highlighted in the structure in **g**. The LRR motifs and ID in the alignment are indicated in colours shown in **g** (the interaction network; right panel). Due to space limitation, the last 9 amino acids in each LRR motif are presented as * in the alignment. For the full alignment, refer to Supplementary Fig. 17a.

The expansion of cell-surface receptors subgroups, including LRR-RLKs, LRR-RLPs, G-lectin-RLKs, and WAK-RLKs, has enabled plants to recognise a broader range of molecules specific to certain stresses and environmental signals. (ii) Development of an increasingly complex signalling network. Cell-surface co-receptors, such as SERKs and SOBIR1 emerged during or around the time of land plant evolution. Moreover, several cell-surface receptors involved in signalling regulation, such as malectin-RLKs (FERONIA)[63] and LRR-RLK-Xa (BIR1, BIR2)[64–67] are found exclusively in Embryophytes. Cell-surface receptors utilise different co-receptors and their cytosolic kinase domains to differentially activate downstream signalling components, including RLCK-VIIs, CDPKs, and MAPKs, which fine-tune the magnitude and specificity of downstream responses[17,68]. Collectively it seems that increasingly intricate and specialised signalling networks enhance the flexibility and regulation of differential responses to keep up with the rapidly changing environment[69]. iii) Adaptation of existing receptors for specific signalling (Fig. 9b). The structural similarities between LRR-RLPs and LRR-RLK-Xbs imply a common origin between immune-specific cell-surface receptors and development-specific cell-surface receptors. The exact nature of the common ancestral form of these receptors, whether an RLK or an RLP, remains, and perhaps will always remain uncertain. Both LRR-RLK-Xbs and LRR-RLPs with ID + 4LRR can be found in land plants, so it is conceivable that LRR-RLK-Xb, with its kinase domain predating land plant emergence, evolved from an integration of an LRR-RLP containing an ID + LRR into an Xb kinase domain[70]. In this scenario, the common ancestral receptors could have recognised a PAMP, with the peptide sequence of this PAMP possibly converted to serve as a phytocytokine to regulate plant developmental processes. Multiple phytocytokines, such as PSK, PSY, SCOOPs, and CLE peptides, are present in plant pathogens and pests[71–75]. Whether the perception of phytocytokines evolved from the perception of PAMPs, or pathogens developed phytocytokine-mimics to repress immune responses remains an open question.

## Methods

### RLK, RLP, and ectodomain identification

We used the same sequences described in the previous publication[4]. We also used the LRR-RLK, LRR-RLP, LysM-RLKs, and Nb-ARC sequences described in the previous publication[4] and did not search them again. The initial set of proteins included only the primary gene models from

all 350 species (12,979,225 proteins in total). Prior to any further HMM searches, sequences were filtered for a minimal length of 250 AA in the case of LRR-RLKs (7,690,505 proteins) or 150 AA in the case of all others (10,224,242 proteins). Based on the presence of a kinase domain (KD) and/or a trans-membrane domain (TM), the proteins fell into three major groups: (1) RLKs with KD and TM, (2) RLPs without KD but with TM, and (3) ectodomain candidates without KD or TM.

To identify RLK candidates, we first searched for the presence of a protein kinase domain (PFAM: PF00069.26) with hmmer (version 3.1b2, option -E 1e-10[76], 439,075 proteins found). If multiple hits were found, only the best match was kept. Potential signalling peptides were removed with SignalP (version 5.0b[77]) to avoid identifying and including signal peptides as TMs (in 139,628 out of the 439,075 candidates). TMs were searched with tmhmm and only candidates with 1–2 TMs were kept, leaving 177,645 proteins (version 2.0[78]). The locations of the KD and the TM were used to split the protein sequence into the endodomain (with KD) and ectodomain (without KD)[4,79].

To identify RLPs and ectodomain candidates (without KD or TM), we first removed all proteins with a kinase domain match (hmmer with the option -E1000), leaving 9,746,585 from the initial 10,224,242 proteins. We next removed potential signalling peptides from the sequences with SignalP because they are sometimes identified as TM domains (trimmed 796,385 sequences). We then searched for TMs with tmhmm and kept proteins with no TM (7,917,087, ectodomains) or 1–2 TM (962,223 RLPs).

Finally, we searched the 177,645 RLK, the 962,223 RLP, and the 7,917,087 ectodomain candidates for the presence of Duf26 (PFAM: PF01657.18), malectin or malectin-like (PFAM: PF11721.9 and PF12819.8), G-type lectin (PFAM: PF01453.25), L-type lectin (PFAM: PF00139.20), and WAK domains: EGF-like (PFAM: PF00008.28), Calcium-binding EGF (PFAM: PF07645.16), and Wall-associated receptor kinase galacturonan-binding (PFAM: PF13947.7) with hmmer (option -E 10). Hits from all searches were combined for each group, and proteins were assigned to the hit with the highest score. Given that we used the previously published LRR-RLKs and LRR-RLPs, all proteins containing LRR repeats were identified with predict-phytolrr and removed from the final sets. For the final number of candidates per species, see Supplementary Data 3.

To test whether RLPs contained potentially functional endodomain sequences, we extracted endo- and ectodomain sequences.

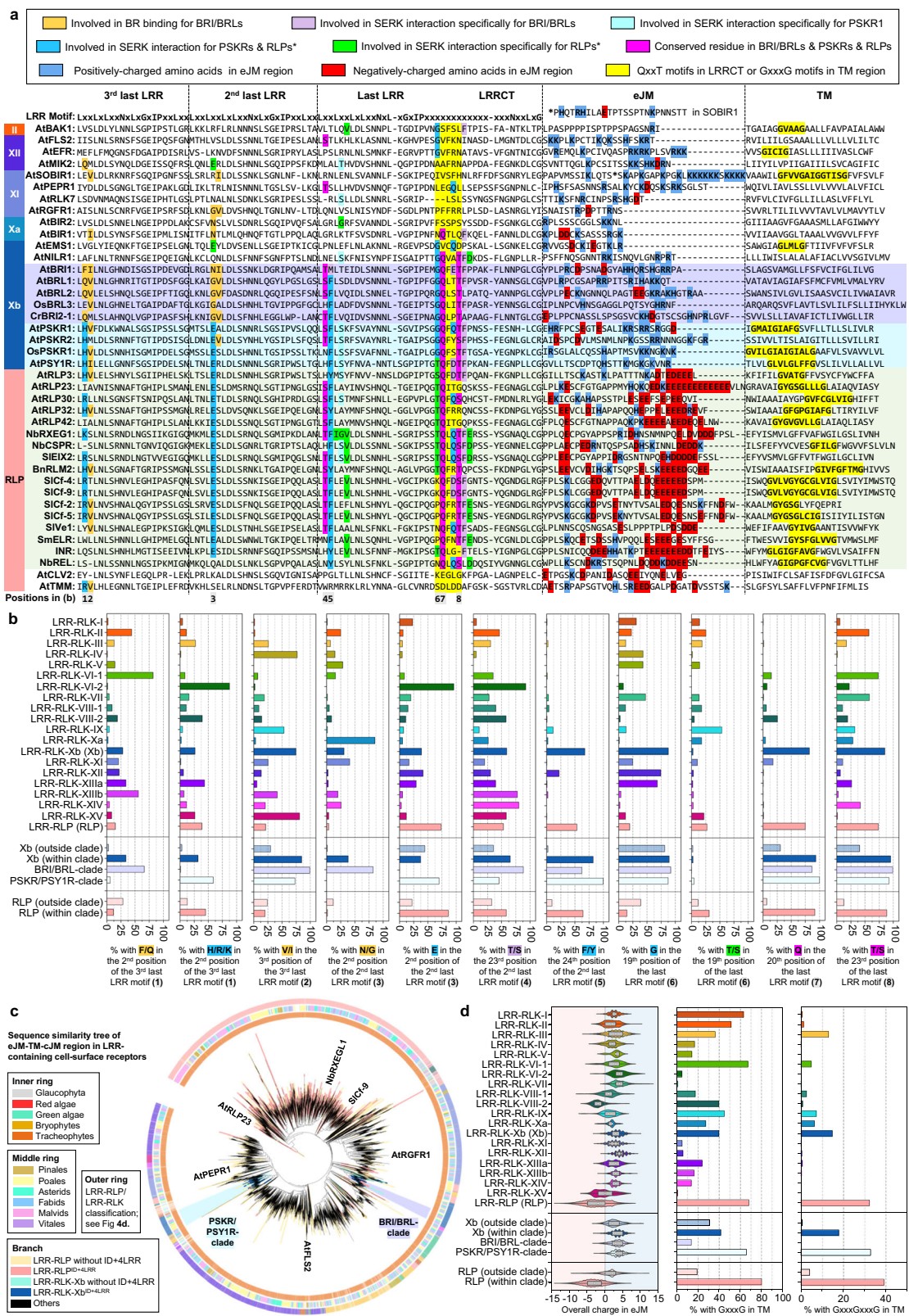

As tmhmm sometimes reverts in- and outside locations, we defined the stretch that was matched by the motif (e.g. WAK domain) as ectodomain sequence. In the case of LRR-RLPs, the ectodomain sequence was defined as the stretch that contained the most LRR repeats. The endodomain was the remaining longest internal sequence stretch. The lengths of these sequences were visualized. We also searched for all available PFAM patterns in the endodomains of all RLPs as well as the

proteins with kinase and TM domain with hmmsearch (with a strict e-value threshold of 1e-10 to ensure a minimum of partial matches). For each endodomain, we only kept the hit with the best score and counted the number of hits per PFAM domain and species. We only kept domains that were found in more than 1 or 1 out of 10,000 input sequences. Within the endodomains of the proteins with kinase and TM domains, 99.42% were best matched by a kinase pattern.

**Fig. 7 | Alignment and features of the terminal four C3, eJM, and TM region in LRR-RLKs and LRR-RLPs. a** Alignment of the C3, eJM, and TM region in LRR-RLKs (from subgroups II, XII, XI, Xa, Xb) and LRR-RLPs. The BRI1/BRL clade is highlighted in purple; PSKR/PSY1R clade is highlighted in cyan and LRR-RLP with the ID + 4LRR clade (see Fig. 4c) is highlighted in light green. Amino acid residues involved in brassinosteroid (BR) binding for BRI/BRL and residues required for SERK interaction for BRI1/BRL, PSKR1, and RXEG1 (LRR-RLP) are highlighted. The colour code for each highlight is indicated in the box on top. For the eJM region, amino acid residues with negative charges are highlighted in red, and amino acids with positive charges are in highlighted blue. The GxxxG motif in the TM region is highlighted in yellow. The interaction sites were calculated using iCn3D[98] with the following thresholds: hydrogen bonds: 4.2 Å; salt bridges/ionic bonds: 6 Å; contacts/interactions: 4 Å. For details, please refer Fig. 5. **b** Percentages of LRR-RLKs or LRR-RLPs with the stated amino acid residues in the corresponding position in **a**. Percentages (%) were calculated by the number of LRR-RLKs or LRR-RLPs in the subgroup with the stated residue divided by the number of LRR-RLKs or LRR-RLPs in the subgroup without the stated residue ×100. **c** Sequence similarity tree of the eJM-TM-cJM region from all LRR-containing cell-surface receptors of 350 species. Branches are colour-labelled as indicated. The inner ring and middle ring indicate the lineage and subclass/order of the corresponding protein (species) from the branch. Outer ring represents the LRR-RLP or LRR-RLK classification, which is indicated in Fig. 4d. Characterized LRR-RLK-Xb and LRR-RLP members are labelled. The BRI/BRL-clade and the PSKR/PSY1R-clades are also labelled. **d** Overall charge distribution in eJM (left), percentage of receptors with GxxxG (middle), and GxxxGxxxG (right) in the TM region. For **b**, **d** RLP/RLK-Xb (outside clade) refers to receptors outside the light grey clade in Fig. 4f. RLP/RLK-Xb (within clade) refers to receptors inside the light grey clade in Fig. 4f. Number of cell-surface receptors (n) in each LRR-RLK subgroup: I, $n = 752$; II, $n = 682$; III, $n = 6572$; IV, $n = 1033$; V, $n = 8$; VI-1, $n = 84$; VI-2, $n = 146$; VII, $n = 1720$; VIII-1, $n = 195$; VIII-2, $n = 411$; IX, $n = 70$; Xa, $n = 96$; Xb, $n = 3182$; XI, $n = 8807$; XII, $n = 12863$; XIIIa, $n = 739$; XIIIb, $n = 465$; XIV, $n = 241$; XV, $n = 548$; Xb (outside clade), $n = 580$; Xb (within clade), $n = 2527$; BRI/BRL clade, $n = 1170$; PSKR/PSY1R clade, $n = 1347$. Number of cell-surface receptors (n) in each LRR-RLP subgroup: LRR-RLP (RLP), $n = 24970$; RLP (outside clade), $n = 5000$; RLP (within clade), $n = 19970$. Boxplot elements: center line, median; bounds of box, 25th and 75th percentiles; whiskers, 1.5 × IQR from 25th and 75th percentiles.

In contrast, in all the RLPs we only found five kinase pattern matches among the 3'860 endodomains of WAK-RLPs (0.13%). In general, endodomains of RLPs were rarely matched by any PFAM pattern: 1.27% of the Duf26-RLPs, 5.03% of the g-type lectin RLPs, 2.1% of the l-type lectin RLPs, and 1.89% of the LRR-RLPs. The WAK-RLPs were slightly different as 15.36% of their ectodomains matched to a PFAM pattern: most of them (13.94%) to a RING finger domain which might be involved in protein-protein interaction (PF13639.7).

### Identification of signalling components

Signalling components were identified using different approaches, but always using primary gene model proteins longer than 150 AA[4]. 150AA cut-off was used to eliminate truncated proteins. For the final number of candidates per species, see Supplementary Data 3.

CNGCs, OSCAs, and RBOHs were identified by hmmer searches (option -E 10) for specific domains. Ion transport protein domains (PFAM: PF00520.32) were used for CNGCs, PHM7_cyt (PFAM: PF14703.7) and RSN1_TM (PFAM: PF13967.7) were used for OSCAs, and FAD-binding domains (PFAM: PF08022.13), ferric reductase like transmembrane components (PFAM: PF01794.20), and ferric reductase NAD binding domains (PFAM: PF08030.13) were used for RBOHs.

EP proteins (EDS1, PAD4, and SAG101) were identified by hmmer searches (option -E 10) for the Lipase (class 3) domain (PFAM: PF01764.26) and the Enhanced disease susceptibility 1 protein EP domain (PFAM: PF18117.2). To further classify the candidates among the known EDS1, PAD4, and SAG101 candidates, we clustered all candidates with MMSeq2 (Release 14-7e284, options --min-seq-id 0.3 -c 0.3[80]). All known EDS1 (*AT3G48090*, *Niben101Scf06720g01024.1*, and *Solyc06g071280*), PAD4 (*AT3G52430*, *Niben101Scf02544g01012.1*, and *Solyc02g032850*), and SAG101 (*AT5G14930*, *Niben101Scf00271g02011.1*, *Niben101Scf01300g01009.1*, *Solyc02g069400*, and *Solyc02g067660*) proteins were found in exactly one cluster each. Hence, we used the three matching clusters as EDS1, PAD4, and SAG101 proteins.

RPW8-NLRs (NRG1 and ADR1) were identified similarly using the NB-ARC (PFAM: PF00931.23) and RPW8 (PFAM: PF05659.12) domains. After clustering as described above, we found all known NRG1 and ADR1 sequences in one single cluster. This allowed us to extract and re-cluster these sequences with more stringent parameters (options --min-seq-id 0.3 -c 0.75). After that, we found all known NRG1 (*AT5G66900*, *AT5G66910*, and *Niben101Scf02118g00018.1*) and ADR1 (*AT1G33560*, *AT4G33300*, *AT5G04720*, and *Niben101Scf02422g02015*) proteins in exactly one cluster each, indicating that the two matching clusters could be considered as NRG1 and ADR1 proteins.

The remaining signalling component candidates (SOBIR1, RLCK-VII, CDPK, MAPK, MAPKK, and MAPKKK) were identified using previously published HMM profiles[81] using hmmer (option -E 10).

The following patterns were used for the families of interest: SOBIR1: RLK-Pelle_LRR-XI-2, RLCK-VII: RLK-Pelle_RLCK-VIIa-1, RLK-Pelle_RLCK-VIIa-2, RLK-Pelle_RLCK-VIIb, CDPK: CAMK_CDPK, MAPK: CMGC_MAPK, MAPKK: STE_STE7, and MAPKKK: STE_STE11.

### Expansion rate of cell-surface receptors and signalling components

The percentages (%) of cell-surface receptors and signalling components from each genome were calculated as (number of identified genes/number of searched genes × 100). Next, the percentages from each species within a lineage (e.g., Rhodophyta or green algae) were grouped and the median percentage was calculated. Median value was used instead of mean to avoid outliers within the lineages. The expansion rate within a species is calculated by ((% cell surface receptors or signalling components in that species)-(median))/(median). For example, the expansion rate of LRR-RLP family in *Marchantia polymorpha* from green algae is calculated by ((%LRR-RLP in *Marchantia polymorpha*)-(median %LRR-RLP in green algae)/(median %LRR-RLP in green algae). Values larger than 0 indicate expansion; values equal to 0 indicate no expansion, and values below 0 indicate contraction. Note that the reliability of the expansion rate is dependent on the number of species used to calculate the median, which is also dependent on the available genomes in Glaucophyta, red algae (Rhodophyta), green algae, and Bryophytes.

### Identification of N-loop outs (NLs) island domains (IDs)

To identify N-loopouts (NLs) or island domains (IDs) in LRR-RLK and LRR-RLP proteins, we used a dataset of previously described LRR-RLKs and LRR-RLPs[4]. We searched the LRR-RLKs again for kinase domains (PFAM: PF00069.26) with hmmer (option -E 1e-10), and kept only the best match for each protein. LRR-motifs and transmembrane domains were searched in both groups with predict-phytolrr[79] and tmhmm[78], respectively. LRR-RLKs were filtered for the presence of internal KD motifs, one or two TM, and at least two external LRR repeats ('internal' was defined as the side with the kinase domain). LRR-RLPs were filtered for the absence of a KD and the presence of one or two TM and at least two external LRR-motifs as defined by the site with more LRR repeats). The outer LRR-motifs were then used to identify NLs and IDs: Individual repeats were grouped into LRR-regions if they were less than 13 AA apart from each other. Gaps between LRR-regions or LRR-motifs that were 15–29 AA or 30–90 AA long were extracted as NL and ID candidates, respectively. After extracting gap sequences, all sequences were again checked for the presence of LRR-motifs using predict-phytolrr and hmmer using all LRR patterns as described previously[4], and LRRsearch[82]. Only NLs and IDs without any LRR match were included in the final dataset.

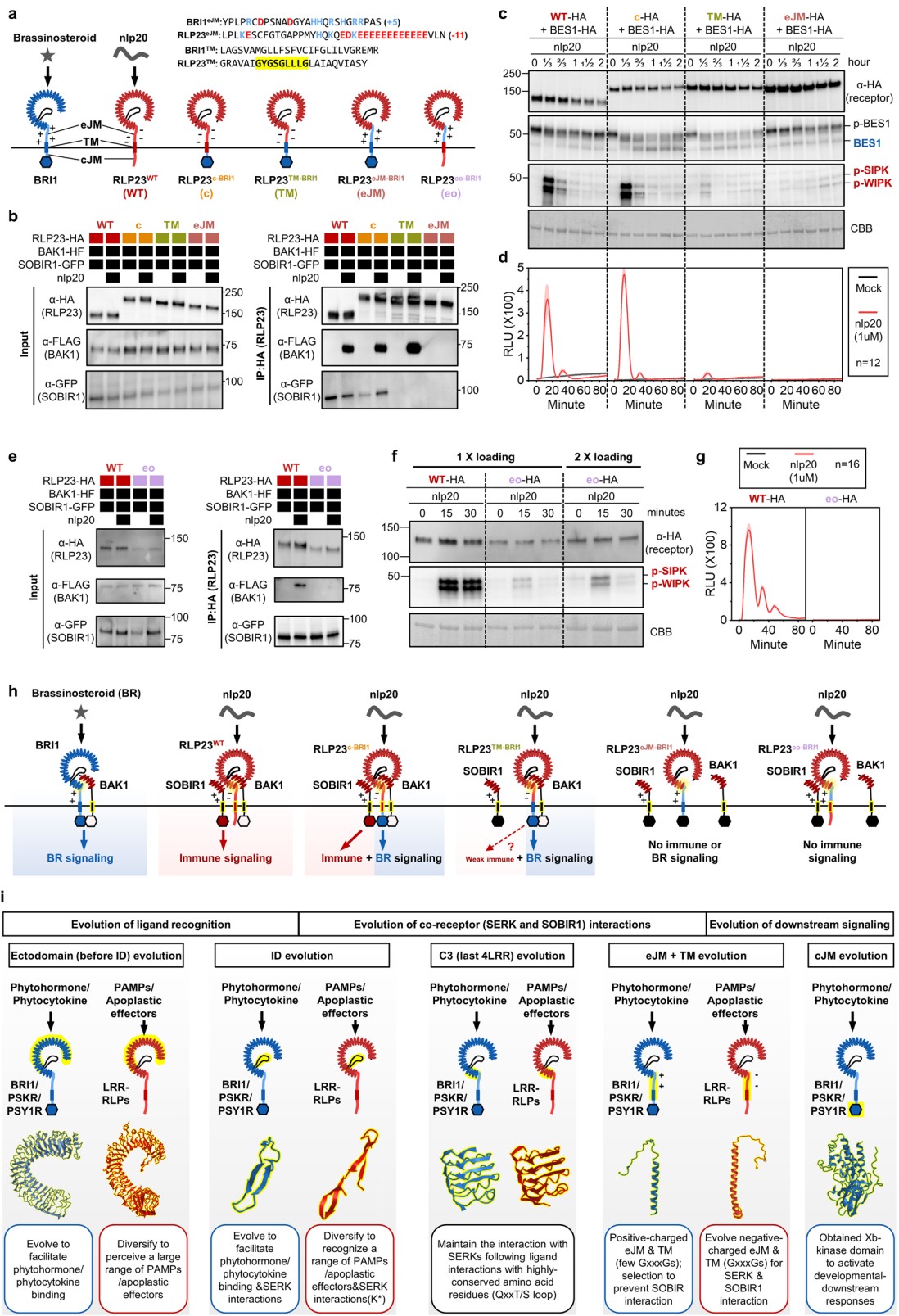

Locations of the NLs and IDs in relation to LRR-motifs and LRR-regions were determined with a custom R-script. ID sequences were aligned to each other with FAMSA[83] without trimming[84] and sequence similarity trees were inferred with FastTree[85] (version 2.1.11 SSE3, option -wag). Trees were rooted with gotree[86] (v0.4.2) using one sequence belonging to the most basal species as outgroup, according to the taxonomic tree. The sequence similarity tree of the IDs was used to cluster the proteins: the tree was converted into a distance matrix using the function cophenetic.phylo() from the R-package 'ape'[87] (version 5.6-2). Distances smaller than 0.2 (*i.e.* less than 0.2 substitutions per site on average) were extracted, converted to similarities, and used as edges in a network. Communities within this similarity network were identified with the function cluster_louvain[88] implemented in the R-package 'igraph'[89] (version 1.2.6).

**Fig. 8 | Adaptation of LRR<sup>ID + 4LRR</sup> to differential downstream signalling pathways. a** Design of AtRLP23-BRI1 chimeras. Different regions (cytosolic, TM+cytosolic, and eJM+TM+cytosolic) of BRI1 were swapped into AtRLP23 as indicated. The alignment of amino acids in the eJM and TM regions from AtRLP23 and AtBRI1 is shown. Amino acid residues with negative charges are in red and amino acids with positive charges are in blue. The GxxxG motif in the TM region is highlighted in yellow. **b**, **e** Immuno-precipitation to test interactions between AtRLP23 chimeras, AtBAK1 and AtSOBIR1. Nb leaves expressing the indicated constructs were treated with either mock or 1 μM nlp20 for 5 min. **c, d, f, g** Functionality testing of AtRLP23 chimeras. Nb leaves expressing the indicated constructs were treated with 1 μM nlp20 and samples were collected at indicated time points. Dephosphorylation of BES1-HA was detected with HA antibody. Phosphorylation of NbSIPK and NbWIPK was detected with p-P42/44 antibody. For **f**, twice the sample of RLP23<sup>oe-BRI1</sup> was loaded as a reference, because RLP23<sup>oe-BRI1</sup> protein accumulation is weaker than that of RLP23<sup>WT</sup>. **d, g** Nb leaf discs expressing the indicated constructs (without BES1-HA) were collected and treated with either mock or 1 μM nlp20 and ROS production was measured for 90 min. For **d** and **g**, solid line, mean; shaded band, s.e.m. RLU,

relative light units. For details of experiental design, refer to the methods section. **h** Schematic model of the interaction between LRR-RLK-Xb<sup>ID+4LRR</sup> and LRR-RLP<sup>ID + 4LRR</sup> with co-receptors to induce differential downstream signalling. Both receptor classes utilise the last 4 LRRs (highlighted in yellow) to interact with SERKs (BAK1). LRR-RLP evolved to interact with SOBIR1 with the GxxxG motifs in TM (highlighted in yellow outline). Coloured hexagons on RLKs indicate activated kinases and black hexagon indicates an inactivated kinase. **i** Modular evolution of different domains in cell-surface receptors to allow diverse ligand recognition and specificity of downstream signalling. Domains or regions that evolved different functions are highlighted in yellow. Bold arrows represent large expansions and diversifications. K* represents the lysine in Kx5Y or Yx8KG motifs in ID from LRR-RLPs. Domain or region structures (from left to right) are obtained from: BRI1 ectodomain (3RGX); RXEG1 ectodomain (7W3X); PSKR1 ID (4Z63); RXEG1 ID (7W3X); PSKR1 C3 (4Z63); RXEG1 C3 (7W3X); PSKR1 eJM-TM (predicted from Alphafold2[20]); RLP23 eJM-TM (predicted from Alphafold2[20]); BRI1 kinase (4OH4). Structures were visualized in iCn3D[98]. For **b, c, e** and **f**, the experiments were repeated at least twice with similar results.

## In-depth phylogeny of the ectodomains and other regions from LRR-RLPs and LRR-RLKs

In-depth analysis of the ectodomains from LRR-RLPs and LRR-RLKs was done using the LRR-RLKs and LRR-RLPs from the NLs and IDs search (see above). We first searched for the C3 domain in each sequence[4,46] with hmmer and selected the best hit. We then pruned the sequences to include everything from the C3 domain to the C terminus. For the LRR-RLKs, we further searched sequences for kinase domains and removed sequences upstream of the start of the kinase domain. That is, for all LRR-RLKs and LRR-RLPs, we extracted regions with C3, eJM, TM, or eJM domains. These sequences were aligned with FAMSA. Specific domains (*e.g.* C3 or eJM) were subsequently extracted from this alignment. After extraction, the sequence similarity trees of specific domains were constructed as described above (FAMSA, FastTree, gotree).

## In-depth phylogeny of the ectodomains from all non-LRR candidates

In-depth analysis of the ectodomains from all Duf26, G-type lectin, L-type lectin, malectin/malectin-like, LysM, and WAK candidates was done using the ectodomain sequences extracted from the ectodomain-only proteins, RLKs, and RLPs. Ectodomain sequences from the ectodomain-only proteins were extracted based on the location of the hmm-pattern match. Phylogenies were constructed as described above with FAMSA, FastTree, and gotree (rooted with a sequence from the most basal species according to the NCBI taxonomy).

## Taxonomic trees

Taxonomic trees used in this study were identical to the ones described previously[4]: The taxonomic tree for visualising the entire data set and selecting outgroups was obtained from NCBI (https://www.ncbi.nlm.nih.gov/Taxonomy/CommonTree/wwwcmt.cgi). The tree used for testing the relationship between the fraction of candidates found and phylogenetic distances, was obtained from a previous report[90]. The latter contained 238 out of the 351 genomes analysed. Sequence similarity trees were visualised and pruned, and figures were generated with iTOL[91].

## Test for similarities in fraction of proteins and phylogenetic relationships

Tests for similarities in the fraction of proteins and phylogenetic relationship were done as described previously[4]: To test whether the fraction of certain proteins found per species correlated with predicted phylogenetic relationships, we converted the fractions and the sequence similarity tree to distance matrices and tested for correlation with mantel tests (R-package vegan, version 2.5-7 with 10,000 permutations). Analogously, we also tested for correlation between distance

matrices obtained for two different sets of proteins. *P*-values were corrected for multiple testing to reflect false discovery rates (FDRs[92]).

## Vector construction

The CDS regions of AtRLP23, AtBRI1, AtPSKR2, AtPSY1R, AtEFR, and AtBES1 were amplified by PCR with KoD one (Toyobo, Japan), and the PCR products were cloned into the epiGreenB5 (3× HA) vector between the *Cla*I and *Bam*HI restriction sites with In-Fusion HD Cloning Kit (Clontech, USA) to generate p35S::BES1-HA or p35S::cell-surface receptor-HA (epiGreenB5-Cauliflower mosaic virus (CaMV) p35S:gene of interest-3 × HA). The constructs were then transformed into *Agrobacterium tumefaciens* strain AGL1 for transient expression in *Nicotiana benthamiana*. All chimeric cell-surface receptors generated in this study contain the EFR signal peptide to ensure consistency between the constructs and expression levels.

## Transient expression in *Nicotiana benthamiana*

*A. tumefaciens* strain AGL1 carrying the binary expression vectors described above were grown on LB agar plates amended with selection antibiotics. Cultures were pelleted, centrifugated, and then resuspended in infiltration buffer (10 mM MgCl₂, 10 mM MES at pH5.6, and 100 μM acetosyringone). The concentration of AGL1 was then adjusted to OD₆₀₀ = 0.5 and syringe-infiltrated into *N. benthamiana* leaves.

## Protein extraction and immunoprecipitation

Protein extraction for immunoprecipitation was performed as previously described[93]. Three days after transient expression, three to four grams of *N. benthamiana* leaves were treated with elicitors and snap-frozen. The tissues were then ground in liquid nitrogen and extracted in extraction buffer (50 mM Tris-HCl at pH 7.5, 150 mM NaCl, 10% glycerol, 5 mM DTT, 2.5 mM NaF, 1 mM Na₂MoO₄•2H₂O, 0.5% poly-vinylpyrrolidone (w/v), 1% Protease Inhibitor Cocktail (P9599; Sigma-Aldrich), 100 μM phenylmethylsulphonyl fluoride and 2% IGEPAL CA-630 (v/v; Sigma-Aldrich), and 2 mM EDTA) at a concentration of 3 mL/g tissue powder. Samples were then incubated at 4 °C for an hour and debris was removed by centrifugation at 13,000 rpm for 10 min at 4 °C. Supernatants were collected, protein concentrations were adjusted to 5 mg/mL, then incubated with rotation for an hour at 4 °C with 50 μL anti-HA magnetic beads (Miltenyi Biotec) for immunoprecipitation. Magnetic beads were then washed twice with extraction buffer and the HA-tagged protein was eluted with sodium dodecyl sulphate (SDS) sample buffer at 95 °C.

## Immunoblotting

Protein extractions were performed as previously described[94]. *N. benthamiana* leaves were infiltrated with elicitors and snap-frozen at

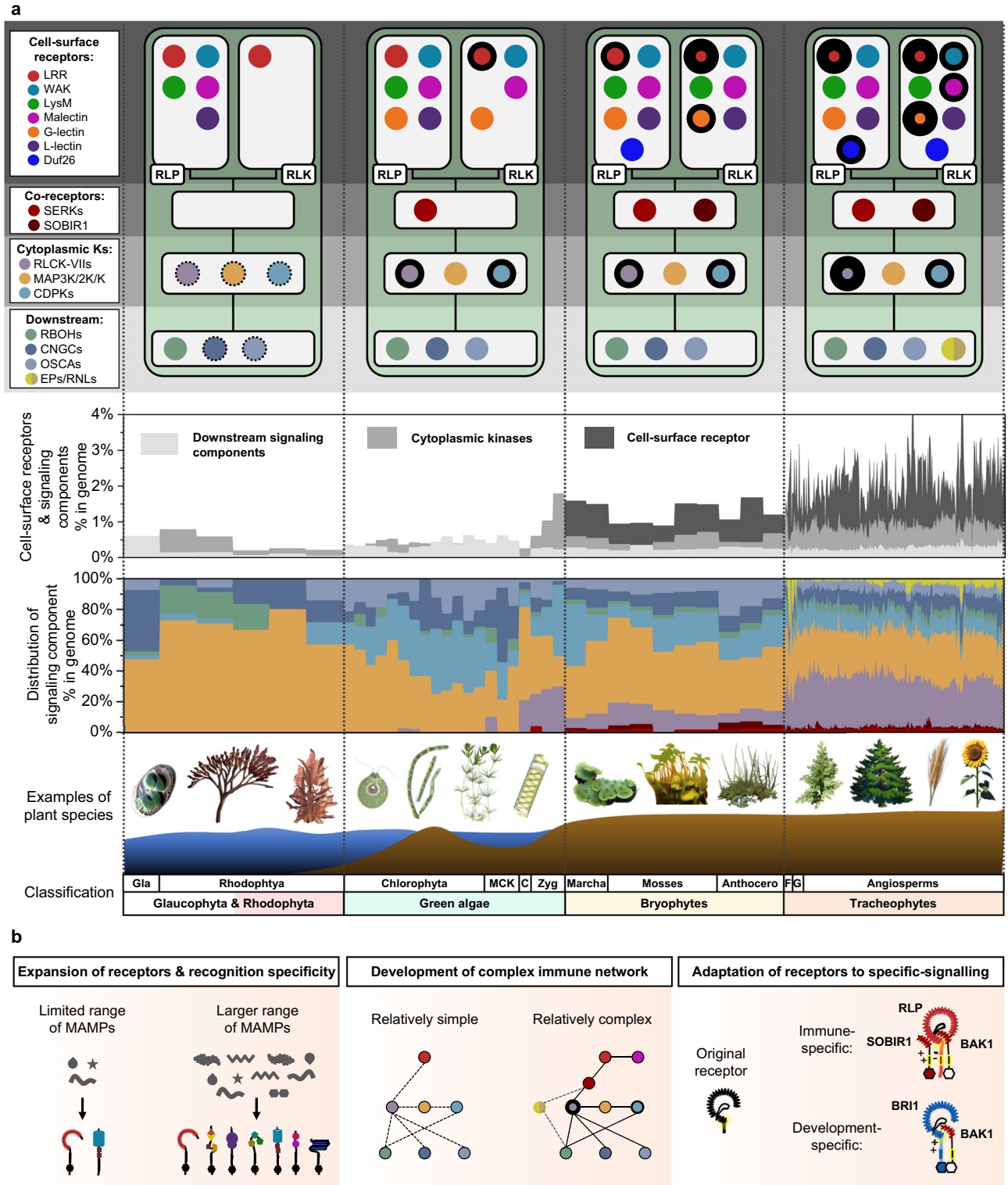

indicated time points. The tissues were then lysed in liquid nitrogen and extracted in 1×NuPAGE™ LDS Sample Buffer (Invitrogen™) with 10 mM DTT at 70 °C for 10 min. Total proteins were then separated by SDS-PAGE and blotted onto a nitrocellulose membrane (Trans-Blot Turbo Transfer System, Bio-Rad). The membrane was then blocked in a solution of either 5% skimmed milk (for BES1 and cell-surface receptor detection) or 5% bovine serum albumin (BSA; for MAPK detection) in Tris-buffered saline, 0.1% Tween 20 detergent (TBST) for an hour. Phosphorylated MAPKs were detected using α-phospho-p44/42 MAPK rabbit monoclonal antibody (D13.14.4E, in 1:2000, Cell Signalling

Technology, USA) in a solution of 5% BSA in TBST overnight at 4 °C. HA-tagged BES1 or cell surface receptors were detected using Anti-HA-Peroxidase, High Affinity, rat IgG₁ antibody (Roche) in a solution of 5% skimmed milk in TBST overnight at 4 °C. For detection of MAPKs, this was followed by incubation with α-rabbit IgG-HRP-conjugated secondary antibodies (1:10,000, Roche, USA) in a solution of 5% BSA in TBST for an hour at room temperature. HRP signal was then detected by Clarity Western ECL Substrate (Bio-Rad) with a LAS 4000 system (GE Healthcare, USA). Nitrocellulose membranes were stained with Coomassie Brilliant Blue (CBB) to ensure equal loading.

**Fig. 9 | The evolutionary trajectory of PTI in plants. a** The top panel depicts the presence of cell-surface receptors, co-receptors, cytoplasmic kinases (cytoplasmic Ks), and downstream signalling components (downstream) in Glaucophyta and Rhodophyta, green algae, Bryophytes, and Tracheophytes. Nodes are labelled in colours as indicated on the left. The absence of a node indicates the absence of a gene family from the lineage. Nodes with dotted outlines indicate the presence of a gene family, but the absence of immunity-related orthologs. Nodes with thick outlines indicate the expansion of gene families. Repeated expansion is indicated with thicker outlines. The middle panel (top) displays the percentages (%) of cell-surface receptors and signalling components in the genome of each species within the lineage. Bars are labelled in colours as indicated. Middle panel (bottom) represents the distribution of signalling components, including co-receptors, cytoplasmic kinases, and downstream signalling components, in the genome of each species within a lineage. Bars are labelled in colours as indicated on top left. The bottom panel shows examples of plant species and the classification of the plant lineages. Gla Glaucophyta, MCK represents Mesostigmatophyceae, Chlorokybophyceae, and Klebsormidiophyceae; C: Charophyta, Zyg Zygnematophyceae, Marcha Marchantiophyta, Mosses Bryophyte, Anthocero Anthocerotophyta, F Lycophytes and Polypodiophyta, G Gymnosperms. **b** Evolution of PTI in plants. Left panel: expansion of PRR family gene repertoires throughout the plant lineage, which leads to recognition of a larger range of PAMPs/MAMPs. Middle panel: PRR co-receptors, EP proteins, and helper NLRs are absent from many algal species. A more complex immune network involving these signalling components apparently developed in vascular plants. Right panel: An ancient PRR with LRR$^{ID}$ + $^{4LRR}$ with unknown function evolved into LRR-RLK-Xbs and LRR-RLPs, which are involved in development- and immune-signalling, respectively. eJM-TM-cJM region of LRR-RLPs evolved to allow interactions with SOBIR1 to induce immunity (negatively charged eJM and GxxxG). LRR-RLK-Xbs utilize Xb kinase domains to induce distinct downstream responses.

### ROS assays

ROS burst assays were performed as described previously[93]. *N. benthamiana* leaf discs were collected with a 4-mm-diameter cork borer and placed in 96-well plates with 120 µl deionised water overnight in the dark (abaxial surface of the leaves facing down). *N. benthamiana* leaf discs were then treated with either mock (water) or 1 µM nlp20 in 20 mM luminol (Wako, Japan) and 0.02 mg ml$^{-1}$ horseradish peroxidase (Sigma-Aldrich). Luminescence was then measured over indicated periods of time with a Tristar2 multimode reader (Berthold Technologies, Germany).

### Reporting summary

Further information on research design is available in the Nature Portfolio Reporting Summary linked to this article.

### Data availability

All data generated or analysed during this study are included in the article or supplementary materials. Proteomes of 350 species used in this study are downloaded from either NCBI, Phyozome13, ensemblplants, JGI, Fernbase, Penium Genome Database, or directly from the publications. A complete list of the proteomes and associated data used in this study is published[4]. Sequences, alignment, and tree files of the identified receptors and downstream signalling components, together with relevant phylogenetic analyses are available on *Zenodo* (https://zenodo.org/records/10059978). Source data are provided in this paper.

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

## Acknowledgements

We thank Yoko Nagai, Naomi Watanabe, and Mizuki Yamamoto for providing technical support, Cyril Zipfel, Simon Snoeck, and Jonathan Jones for discussions and critical reading of the manuscript. We also thank Markus Albert for sharing the information on chimeric receptors and BES1-HA experiments. B.P.M.N. is an International Research Fellow of the Japan Society for the Promotion of Science (Postdoctoral Fellowships for Research in Japan (Standard), 21F21793). The research was financially supported by MEXT/JSPS KAKENHI Grant Numbers, 20H02994, 21K19128, JPJSBP1-20193222 (to Y.K.), 20H05909 and 22H00364 (to K.S.).

## Author contributions

B.P.M.N., Y.K., and K.S. conceived and conceptualised the study; B.P.M.N., Y.K., M.W.S., and M.W. designed the bioinformatic analyses; M.W.S. and M.W. performed the bioinformatic analyses; B.P.M.N. performed the vector construction with assistance from Y.K. B.P.M.N. performed protein extractions, immunoprecipitation, immunoblotting, and ROS assays; B.P.M.N. wrote the original draft; and B.P.M.N., M.W.S., M.W., Y.K., and K.S. reviewed and edited the manuscript.

## Competing interests

The authors declare no competing interests.
