## [Peer Review File · Nature Communications]

Evolutionary Trajectory of Pattern Recognition Receptors in PlantsReviewer #1 (Remarks to the Author):

In this study, Ngou et al. studied the evolutionary trajectory of pattern recognition receptors in plants. This reviewer appreciates the great efforts. However, this reviewer thinks that there are major technical flaws in this manuscript, which is crucial for most of the conclusions. Moreover, some key conclusions have been explored by previous studies, which were neglected by this manuscript.

Major:

1. In general, it is highly challenging to identify RLPs. In brief, the authors identified RLPs by searching ectodomains and TM domains (plus and no kinase domain). However, this reviewer does not think a protein with an ectodomain plus 1 to 2 TM domains is necessarily an RLP. It is possible that this protein might be derived from a protein with the ectodomain, TM, and other domains. Phylogenetic analyses of ectodomains might represent a possible strategy to identify RLPs, which should cluster together with (either within or basal to) RLKs. On the other hand, it is not reasonable to perform phylogenetic analysis for some ectodomains, especially LRRs (it does not make any sense to align, for example, 4 LRR repeats with 5 LRR repeats, for example). Also, it is unclear whether C3-F domain is only present in RLKs.

2. Similarly, a protein with ectodomain, TM, and kinase domains is not necessarily an RLK. Shiu et al. suggest that RLKs form a monophyletic gene family (PMID: 11526204). RLK-like architecture (especially for LRR-RLKs) might have originated independently. Indeed, the NBS-LRR architectures of plant and animal NLRs evolved in independent events (PMID: 28096345). A re-analysis of kinase domains is necessary.

3. Inadequate citation. Line 31: "However, the origin of PRRs in plants remains unclear." This reviewer does not agree with this crucial statement. Previous studies, such as Shiu et al. (2001; PMID: 11526204) and Gong et al. (2021; PMID: 33423360), have dissected the origin and evolutionary trajectory of RLKs. In terms of RLK origin, this study does not provide much advancement. These studies were just neglected in this manuscript. Many conclusions in this manuscript are covered by previous studies.

4. Line 53-55: "Except for LRR-RLPs, all six families of RLP are basal to the RLK families This suggests the intriguing possibility that numerous RLKs may have evolved directly from RLPs through the integration of kinase domains". This reviewer does not think these conclusions are supported by phylogenetic analyses. Distribution itself cannot tell these conclusions. Once again, phylogenetic analyses of ectodomains are necessary (of course, sequences from non-RLP/RLK should be included, and roots should be specified).

5. Line 117: "we aligned the last four LRR motifs (referred to as the C3 region) from LRR-RLKs and LRR-RLPs." LRR regions are very complex. It is not necessary these LRR motifs are derived from a single source (the prerequisite for meaningful alignment). For any conclusions, branch supports should be mentioned.

6. Phylogenetic analysis of JM-TM-cJM does not make sense, because they are not necessarily homologous, and they are extremely short in length.

7. Line 46-47. The ectodomains studied do not cover all the known RLK-associated ectodomains.

Minor:

Line30-31: The statement "PTI is conserved in animals, plants..." is misleading. The mechanisms are not that conserved.

Figure 1-2: Figures 1 and 2 are difficult to follow (much like models). This reviewer think it is better to re-organize readable figures and key numbers should be included.

Figure 3: The key genes studied should be highlighted in panel C.

Reviewer #2 (Remarks to the Author):

Structural knowledge of plant receptors has greatly advanced in recent years, but empirical structural data is necessarily confined to single proteins from a limited number of species. Phylogenomic analysis is highly complementary to a structural approach, especially by placing regions essential for protein function in the context of sequence conservation and divergence across hundreds of species. The manuscript by Ngou et al is at the cutting edge of this type of combined analysis -- "structural phylogenomics" -- and uses the toolkit to analyze LRR-type cell surface receptors, a gene family of broad interest to plant developmental biologists and immunologists

A first major strength of the work is to set new standards in the comparative analysis of key regions of receptor ectodomains. This is the first work in my knowledge to definitively relate structural aspects of BAK1/SERK association to conservation of key binding residues in both LRR-RLK and LRR-RLP families. Critical facts about both structure and evolutionary conservation are made accessible through clear figures and an extensive supplement. Similarly, the treatment of island domain and transmembrane regions is rigorous and complete (although see my Concern 1 below about island domain motif analyses). Combined, this work brings the classic, seminal analyses of LRR-RLK/RLP ectodomain architecture (Shiu 2001, Fritz-Laylin 2005) up to date in the genomic era.

A second major contribution is important functional insights into LRR-RLK-Xb and LRR-RLPs using chimeric RLP23 constructs and specific immunity and development signaling outputs. The finding of C3 functional conservation, but eJD-TM-KD functional divergence, has broad implications for cell surface receptor biology. It also lends functional support to the striking notion that Xb and RLP subfamilies may share a common evolutionary origin. Since previous analyses have generally focused only on kinase domains rather than LRR domain C3, they have missed this incredible, fundamental observation relating RLKs and RLPs.

I have three major concerns:

1. The phylogenomic analysis of LRR-RLP island domains is not as well developed as TM and C3 regions. Interestingly, the ID section has been truncated relative to the Biorxiv preprint. (In fact, Extended Fig. 9b and Suppl. Fig 33 are now no longer referenced in the text.) I believe the authors have likely held back as not to make a claim about the common ancestry of RLK-Xb and RLP. However, the useful observation of Kx5Y and Yx8KG motifs as distinguishing features of LRR-RLP IDs is now lost.

To restore this aspect of the paper, the authors should restore the missing Snoeck et al reference (<https://doi.org/10.1016/j.pmpp.2023.102004>) and restore the important sentence from the preprint, which can be inserted around line 174: "More than 75% of IDs from LRR-RLPs contain at least one of two lysine motifs, Kx5Y or Yx8KG, while less than 5% of IDs from LRR-RLK-Xb have either motif (Extended Figure 9b, Supplementary figure 33)." I feel that even just a short, motif level analysis of IDs would help match the comprehensive analysis of the C3 domain.

2. Fig 4e demonstrates that RLP23 chimeras with BRI1 KD or TM-KD can induce BES1 de-phosphorylation. However, treatment with brassinolide (BL) as a positive control for BES1 bandshift is missing, and so the magnitude of de-phosphorylation (partial after 1 hr) cannot be compared to a full strength BRI1-mediated response. I expect BL would work through endogenous NbBRI1, but perhaps would require BRI1 overexpression. Brassinolide-induced responses are not normally studied in transient expression assays, and therefore I feel BL treatment is an important control not just for the current conclusion about RLP23-c-BRI1 function, but for establishing N.benthamiana transient expression as a system for RK signaling crosstalk.

3. In data and materials availability, the authors state that "Sequences of the identified receptors and phylogenetic analyses will be uploaded and available to the public upon publication." This is a critical component for peer review. Please make these available through Zenodo or other repository as part of the revised manuscript.

Minor points:

- Fig 1: It is not clear from the legend what the individual points represent on the expansion rate boxplots. Based on the text, my guess is a single species in the derived group vs a tge median percentage in the early diverging group? How many pairwise comparisons are represented in each panel? What is the x-axis unit?
- Fig 2+3: the term "PRR" is used in panel a, but throughout the text the authors are careful not to assume pattern recognition functions. I suggest removing "PRR" from the figures.
- Line 107 and Fig 3a: I can't tell if the black borders between pink and blue segments are thick dividing lines or separate RLK categories. Since only blue and pink sections are visible in the outer ring, how is this consistent with the "94.7%" claim in line 107?
- line 125: should refer to Extended figure 7, not 6
- line 164: "absence of a specific T residue before the Qxxx motif in RLP23" -- this phrasing makes it sound like RLP23 lacks the Thr. I suggest to just remove the words "in RLP23"
- Fig 4c: using 6-pointed Star of David (a religious symbol) for brassinosteroid is distracting. Is a 5-point star OK?
- The title feels overly broad, like a review rather than a primary data article. It is hard to tell from the current title how it differs from the previous analysis by Ngou et al 2022, Nature Plants, "Concerted expansion and contraction of immune receptor gene repertoires in plant genomes". I would suggest using the present title to draw attention to the RK-Xb and LRR-RLP findings.(something like "Functional conservation of LRR-associated regions between receptor kinases and receptor-like proteins")

Reviewer #3 (Remarks to the Author):

The manuscript by Ngou et al. discussed the shared origin and function of cell surface receptors across the plant kingdom. In the first part of the manuscript, Ngou et al. expand on a previous curation of LRR cell surface receptors published last year to include additional ectodomains, and investigate the origin of the different receptor families along plant evolution, identifying the origin and expansion of different ectodomain-containing receptors in RLP and RLK forms. They find that the ectodomains themselves mostly exist already in Glaucophyta (except G-lectin and DUF26) and that the RLP form is usually ancestral to the RLK form. From that point, the paper focuses on LRR RLPs and RLKs and investigate their common origin, diverging roles and mechanistic basis for this divergence. At first, the authors focus on structural features in the LRR repeats, the C3, NL and ID motifs. They show that in LRR receptors containing IDs, RLPs and RLKs clade together in a phylogenetic analysis based on the C3 region, and conclude that these RLPs and RLKs share a common origin. They then show that a conserved motif in the C3 region is responsible for BAK coreceptor recruitment upon ligand binding using protein chimera based on RLP23. They then use the RLP23 LRR domain to generate chimeras in order to investigate the roles of the TM and surrounding regions in coreceptor recruitment and the roles of the kinase domain presence or absence in activation of developmental and immunological responses. The authors demonstrate that these chimeras are functional and define the protein regions responsible for recruitment and signaling, supporting the hypothesis that theses various receptors share a common origin and underwent evolutionary diversification illustrated in Extended Data figure 10b to fulfill their diverse roles in immunity and development.

The paper presents a fascinating story of evolution, combines evolutionary analysis with functional protein assays to support the hypotheses, and includes multiple useful datasets. While the general concept of a the interchangeability of ectodomains, signaling mechanisms and biological processes is not novel, the scale of the analysis and the elegant demonstration of the concepts are valuable, impressive and important for the advancement of our understanding of the topic. This paper is of great interest to the plant immunity field, but also to the fields of protein evolution and signal transduction. The analyses are in depth and comprehensive, and the results are generally well presented. I enjoyed reading the paper in depth, and believe many will find it interesting and useful.

Comments:

- One general criticism is that the paper is very long and contains a very large number of extended and supplementary data. Probably as a result of iterations in the revision process, the focus in the main text is not well reflected in the choice of the main figures. Main figure 2 is only superficially discussed in a single paragraph, while data in extended figure 1 for example is crucial for understanding but omitted from the main text. I would suggest rethinking what goes to the main figures and what goes to extended in light of the final version of the text presents. In hindsight, this work might have been more readable as two independent papers but this is of course at the discretion of the authors.
- Analysis of protein sequences derived from such a large number of genomes is prone to gene model annotation errors. This holds special importance as the analysis performed by the authors is searching for absence of domains and intervening sequences – both can be detected as a result of annotation errors of intron-exon junctions. While this is inherent to working with multiple complex eukaryotic genomes, some simple additional analyses could be added that will make the reader more confident.
- Such analyses should include: When only the ectodomain is detected, is it associated with another domain? Are those NLR genes? In genes annotated as RLPs, is there indeed only a short intracellular part to the protein or could they encode for an additional intracellular domain? Can this unannotated domain be a kinase, changing the picture regarding RLPs and RLKs? How many of the IDs and NLRs detected are homologous to such sequences from close species?
- The use of a single PFAM annotation for kinase domain detection might cause false negative results for protein RLK detected as RLPs when the kinase is misannotated. Incorporation of additional kinase PFAM such as PF07714 or analysis of unannotated domains of receptors with more sensitive tools such as HHpred could support the results. As absence of a detected kinase domain is what makes a protein an RLP in this work, and as the claims regarding the evolutionary relation between RLPs and RLKs are central here, I believe a more careful analysis is merited to give the readers a more reliable picture. This can also be relevant to the discussion of the antiquity of the kinase domain at the end of paragraph about origin and expansion of cell-surface receptors.
- The description of expansion and contraction of gene families in the main text is lacking, and is only well explained in the supplementary text. A clear explanation that % enrichment is in relation to the total number of genes in the genome (and not to the total number of PRRs for example) is required. Such analysis is also prone to artifacts as the expansion of another gene family in the genome would result in apparent contraction of the other gene families. This should be presented and discussed in the main text.
- “Moreover, downstream signalling components exhibit limited expansions compared to cell-surface receptors (Figure 2).” – what about the expansions of RLCK and CDPK? Are they considered non-significant?
- Regarding Figure 3 - Are there results for co-IP of the PSKR2 chimera with the coreceptors? As the text refers to the inability of this chimera in inducing immune response and the importance of the T residue, these results are relevant.
- The paragraph introducing the concepts of NLRs and IDs is not referenced at all, preventing the reader from understanding what was known and what is discovered by this work. As a side note, the use of the acronym ID that is already used for “integrated domains” in NLR immune receptors is unfortunate, but not the fault of the authors.
- In the method section, it is not clear enough how datasets were generated – many references to “included” or “removed” when it is not clear enough what dataset was modified. Please specify exactly the proteomic dataset scanned, the search done, the number of hits identified, and refrain from basing the methods for this search on “as previously described”. Please provide final lists of protein accessions and sequences so the amazing dataset you created can be used by others, as beautifully done in previous works by the first author. Raw data for each of the trees and lists should be provided.
- References in the main text to Extended Data Figure 6 actually point to Ext. Data Fig. 7
- The graphical representation of the NLR in Ext.Data Fig 5 is very hard to see. Highlighting it would be useful.
- While in the C3 phylogenetic tree in Fig. 3 it is clear that RLPs and RLKs are in sister clades, this is not so easily identifiable in the tree of Ext.Data Fig 9. Do the authors refer to the fact that the pink and blue leaves are intermixed in the same clades? It is hard to see visually. Better description of it in the text would be useful if the authors are making the claim that they are evolutionarily related.

Reviewer #4 (Remarks to the Author):

The study presents a comprehensive comparative genomics analysis of RKL and RLP evolution by examining general multi-domain protein architectures. There are several interesting, focused analyses, including emergence and co-evolution of signaling co-receptors that is corroborated with molecular biology experiments. Additional comparative analyses are focused on motifs and their biochemical properties.

I think that the study would be greatly improved by statistical analyses, putting in context of other global RLK/RLP analyses as well as including structure-based interpretation of evolutionary and biochemical data. Please, see a list of suggestions and methodologies below:

- Protein family expansion and contraction are currently done with domain counting in organisms that are currently present. These are very general analyses that would be appropriate for a review but need to be done more robustly for a research paper to support presented conclusions. There are standard protein family evolutionary analyses quantifications that allow to model expansion and contraction based on comparing species phylogenies (single gene copy ortholog BUSCO genes) to protein family phylogenies. I think it would be essential to implement these. For suggested methods details, please see following paper for standard methodology:

o Shao ZQ, Xue JY, Wu P, Zhang YM, Wu Y, Hang YY, Wang B, Chen JQ. Large-Scale Analyses of Angiosperm Nucleotide-Binding Site-Leucine-Rich Repeat Genes Reveal Three Anciently Diverged Classes with Distinct Evolutionary Patterns. *Plant Physiol.* 2016 Apr;170(4):2095-109. doi: 10.1104/pp.15.01487. Epub 2016 Feb 2. PMID: 26839128; PMCID: PMC4825152.

o De Bie T, Cristianini N, Demuth JP, & Hahn MW (2006) CAFE: a computational tool for the study of gene family evolution. *Bioinformatics* 22(10):1269-1271.

o Hahn MW, De Bie T, Stajich JE, Nguyen C, & Cristianini N (2005) Estimating the tempo and mode of gene family evolution from comparative genomic data. *Genome research* 15(8):1153-1160.

- There have been several previous publications on global RLK/RLP analyses that authors should take into account and compare results to:

o Man J, Gallagher JP, Bartlett M. Structural evolution drives diversification of the large LRR-RLK gene family. *New Phytol.* 2020 Jun;226(5):1492-1505. doi: 10.1111/nph.16455. Epub 2020 Feb 29. PMID: 31990988; PMCID: PMC7318236.

o Simon Snoeck Bradley W Abramson Anthony GK Garcia Ashley N Egan Todd P Michael Adam D Steinbrenner (2022) Evolutionary gain and loss of a plant pattern-recognition receptor for HAMP recognition *eLife* 11:e81050.

<https://doi.org/10.7554/eLife.81050>

- Motif enrichment analyses need to be statistically quantified, either by cumulative hypergeometric or cumulative binomial distributions.

- There has been a breakthrough in structure modeling with AlphaFold2 so models for most genes in model species are available. Could you interpret evolutionary plus the biochemical data in structure context?

Respond to reviewer's comments

Reviewer #1 (Remarks to the Author):

In this study, Ngou et al. studied the evolutionary trajectory of pattern recognition receptors in plants. This reviewer appreciates the great efforts. However, this reviewer thinks that there are major technical flaws in this manuscript, which is crucial for most of the conclusions. Moreover, some key conclusions have been explored by previous studies, which were neglected by this manuscript.

Response to reviewers: We sincerely appreciate the valuable comments provided by reviewer 1, which have greatly contributed to improving the quality of our manuscript. We also want to apologize for not citing important previous papers. We missed them as we tried to adhere to the manuscript's length limitations. We have carefully considered each comment and have made the necessary revisions as shown below. We believe that these revisions adequately address reviewer 1's comments.

Major:

1. In general, it is highly challenging to identify RLPs. In brief, the authors identified RLPs by searching ectodomains and TM domains (plus and no kinase domain). However, this reviewer does not think a protein with an ectodomain plus 1 to 2 TM domains is necessarily an RLP. It is possible that this protein might be derived from a protein with the ectodomain, TM, and other domains. Phylogenetic analyses of ectodomains might represent a possible strategy to identify RLPs, which should cluster together with (either within or basal to) RLKs. On the other hand, it is not reasonable to perform phylogenetic analysis for some ectodomains, especially LRRs (it does not make any sense to align, for example, 4 LRR repeats with 5 LRR repeats, for example). Also, it is unclear whether C3-F domain is only present in RLKs.

2. Similarly, a protein with ectodomain, TM, and kinase domains is not necessarily an RLK. Shiu et al. suggest that RLKs form a monophyletic gene family (PMID: 11526204). RLK-like architecture (especially for LRR-RLKs) might have originated independently. Indeed, the NBS-LRR architectures of plant and animal NLRs evolved in independent events (PMID: 28096345). A re-analysis of kinase domains is necessary.

Response to reviewers: Many thanks for the comments. First of all, we defined a protein with an ectodomain plus 1/2 TM domains with the absence of kinase domain as a receptor-like protein (RLP). Of course, it is possible that the cytosolic region of some RLPs contains other domains. We have checked the AA length of the intracellular part/domain in RLKs and RLPs (see supplementary data 1). While RLKs have much longer intracellular domains (around 300-400 AAs), RLPs mostly have very short intracellular domains (less than 100 AAs). Furthermore, we took the intracellular domains of the RLPs and performed PFAM search on them (see supplementary data 1). We failed to see any significant enrichment of any motifs, except for RING-finger domain in WAK-RLPs. Overall, we believe that the cytosolic region in RLPs is mostly short and does not contain functional motifs or domains. Nevertheless, we believe that these proteins would still be classified as RLPs, since our definition for RLPs is that the cytosolic region does not contain a kinase domain (that scores above the search threshold). We do not define an RLP as a protein with non-functional cytosolic motifs of any sort. We have responded to and explained the reason to align part of the LRR region below (see response to point 5 and 6). C3-F region is present only in receptors that contain LRR and a transmembrane domain, thus they are not present only in LRR-RLKs, but also in LRR-RLPs.

Similar to our response to the identification of RLPs, we defined a protein with an ectodomain, 1-2 TM domains, and kinase domain as a receptor-like kinase (RLK). We understand that Shiu et al. suggested that RLKs form a monophyletic gene family. However, we believe that even if an RLK contains a kinase domain outside of this monophyletic gene family, it is still possible that this RLK can activate downstream response through its kinase domain. We do believe that these RLK-like/RLP architectures can originate independently (see response to point 4). However, this does not change our definition of an RLK (ectodomain+TM+kinase).

3. Inadequate citation. Line 31: “However, the origin of PRRs in plants remains unclear.” This reviewer does not agree with this crucial statement. Previous studies, such as Shiu et al. (2001; PMID: 11526204) and Gong et al. (2021; PMID: 33423360), have dissected the origin and evolutionary trajectory of RLKs. In terms of RLK origin, this study does not provide much advancement. These studies were just neglected in this manuscript. Many conclusions in this manuscript are covered by previous studies.

Response to reviewers: Many thanks for the comment. PRRs are involved in PAMP recognition. Shiu et al. and Gong et al. have indeed dissected the origin and evolutionary trajectory of RLKs in the plant lineage. However, the exact origin of immune-specific PRR families (such as the LRR-RLPs) remains unsolved. We have updated this statement in the introduction. We did not claim to have provided advancement in the origin of RLKs. We have now incorporated and cited these two papers in the introduction.

4. Line 53-55: “Except for LRR-RLPs, all six families of RLP are basal to the RLK families This suggests the intriguing possibility that numerous RLKs may have evolved directly from RLPs through the integration of kinase domains”. This reviewer does not think these conclusions are supported by phylogenetic analyses. Distribution itself cannot tell these conclusions. Once again, phylogenetic analyses of ectodomains are necessary (of course, sequences from non-RLP/RLK should be included, and roots should be specified).

Response to reviewers: Many thanks for the suggestions. We have now aligned the G-lectin-, WAK-, Duf26-, L-lectin-, LysM- and Malectin-ectodomains from the ectodomain-only proteins, RLPs, and RLKs within either a subset of 25 species or all 350 species (see supplementary figure 4). In some cases, we can clearly see RLKs that are likely to have evolved from RLPs (WAK-RLK, for example). While in other cases (such as the LysM-RLKs), it is unclear whether they evolved from ectodomain-only proteins or RLPs. Thus, we concluded in the main text that ‘RLKs emerged from either the RLPs (through integration of kinase domains) or directly from ectodomain-only proteins (through integration of TM+KD)’. Thanks again for highlighting this important point.

5. Line 117: “we aligned the last four LRR motifs (referred to as the C3 region) from LRR-RLKs and LRR-RLPs.” LRR regions are very complex. It is not necessary these LRR motifs are derived from a single source (the prerequisite for meaningful alignment). For any conclusions, branch supports should be mentioned.

Response to reviewers: Many thanks for the suggestions. We agree that the LRR region is very complex and that it’s not necessary these LRR motifs are derived from a single source. However, the goal/point to generate the C3 region phylogenetic tree is to show that the LRR-RLPs and LRR-RLK-Xbs have C3 regions that are closely-related (or highly similar) to each other, as we have further shown with biochemistry data. Nevertheless, we believe that the alignment of the AA sequences in this region helped us to show that the LRR-RLPs and LRR-RLP-Xbs C3 region are similar to each other, thus it is meaningful and provided clues to the evolutionary origin of these two classes of receptors. There are too many branches on the phylogenetic trees to show the branch supports individually. We have uploaded the trees on *Zenodo* (<https://zenodo.org/records/10059978>) for the readers and reviewers.

6. Phylogenetic analysis of JM-TM-cJM does not make sense, because they are not necessarily homologous, and they are extremely short in length.

Response to reviewers: Many thanks for the suggestions. Similar to our answer to point 5, we agree that the JM-TM-cJM region are not necessarily homologous. However, the goal/point to generate this phylogenetic tree is to show that the LRR-RLPs and LRR-RLK-Xbs have very distinct JM-TM-cJM region, as further shown in figure 6a. It is because of the very distinct AA and biochemical properties of JM-TM-cJM region of LRR-RLPs and LRR-RLK-Xbs that allows them to specialize in two distinct downstream signalling pathway. Thus, we think that it is meaningful to show these data, as they provided clues to how these two classes of receptors diverged to activate two distant signalling pathways.

In a more general sense, it is unclear to us how we should decide which sequences are non-homologous if we don't even compare them to each other. Indeed, for a subset of "closely" species, one could identify high confidence homologous genes through synteny analysis, but that's not possible for all of them because the synteny breaks down. Also, addressing synteny in 350 genomes, from which not all are chromosome-level assemblies seems like a project on its own. Hence, building a phylogenetic tree based on sequence similarity of a given domain to test whether domains from two groups are found in separate branches or not seems to be the only way to show whether they are potentially having the same origin or not. Nevertheless, we have changed 'phylogenetic tree' into 'sequence similarity tree' in the manuscript to be more technically correct with our claims.

7. Line 46-47. The ectodomains studied do not cover all the known RLK-associated ectodomains.

Response to reviewers: Many thanks for the suggestions. We focused on LRR-, G-lectin-, WAK-, Duf26-, L-lectin-, LysM- and Malectin-containing receptors in this manuscript due to their characterised roles in plant immunity. We have stated in the manuscript with references that there are additional RLK families with others ectodomains.

Minor:

Line30-31: The statement "PTI is conserved in animals, plants..." is misleading. The mechanisms are not that conserved.

Response to reviewers: Many thanks for the suggestions. While the downstream immune activation mechanism of pattern-triggered immunity is not conserved in different kingdoms. The mechanism/concept of pathogen-associated molecular patterns (PAMP) perception is conserved. We have rewritten this statement to avoid confusion.

Figure 1-2: Figures 1 and 2 are difficult to follow (much like models). This reviewer think it is better to re-organize readable figures and key numbers should be included.

Response to reviewers: Many thanks for the suggestions. We have re-organize the figures (now 9 main figures) and main text to help the readers to navigate the manuscript.

Figure 3: The key genes studied should be highlighted in panel C.

Response to reviewers: Many thanks for the suggestions. We have labelled some key genes in that figure.

Reviewer #2 (Remarks to the Author):

Structural knowledge of plant receptors has greatly advanced in recent years, but empirical structural data is necessarily confined to single proteins from a limited number of species. Phylogenomic analysis is highly complementary to a structural approach, especially by placing regions essential for protein function in the context of sequence conservation and divergence across hundreds of species. The manuscript by Ngou et al is at the cutting edge of this type of combined analysis -- "structural phylogenomics" -- and uses the toolkit to analyze LRR-type cell surface receptors, a gene family of broad interest to plant developmental biologists and immunologists

A first major strength of the work is to set new standards in the comparative analysis of key regions of receptor ectodomains. This is the first work in my knowledge to definitively relate structural aspects of BAK1/SERK association to conservation of key binding residues in both LRR-RLK and LRR-RLP families. Critical facts about both structure and evolutionary conservation are made accessible through clear figures and an extensive supplement. Similarly, the treatment of island domain and transmembrane regions is rigorous and complete (although see my Concern 1 below about island domain motif analyses). Combined, this work brings the classic, seminal analyses of LRR-RLK/RLP ectodomain architecture (Shiu 2001, Fritz-Laylin 2005) up to date in the genomic era.

A second major contribution is important functional insights into LRR-RLK-Xb and LRR-RLPs using chimeric RLP23 constructs and specific immunity and development signaling outputs. The finding of C3 functional conservation, but eJD-TM-KD functional divergence, has broad implications for cell surface receptor biology. It also lends functional support to the striking notion that Xb and RLP subfamilies may share a common evolutionary origin. Since previous analyses have generally focused only on kinase domains rather than LRR domain C3, they have missed this incredible, fundamental observation relating RLKs and RLPs.

Response to reviewers: We sincerely appreciate the valuable comments provided by the reviewer 2, which have greatly contributed to improving the quality of our manuscript.

I have three major concerns:

1. The phylogenomic analysis of LRR-RLP island domains is not as well developed as TM and C3 regions. Interestingly, the ID section has been truncated relative to the Biorxiv preprint. (In fact, Extended Fig. 9b and Suppl. Fig 33 are now no longer referenced in the text.) I believe the authors have likely held back as not to make a claim about the common ancestry of RLK-Xb and RLP. However, the useful observation of Kx5Y and Yx8KG motifs as distinguishing features of LRR-RLP IDs is now lost.

To restore this aspect of the paper, the authors should restore the missing Snoeck et al reference (<https://doi.org/10.1016/j.pmpp.2023.102004>) and restore the important sentence from the preprint, which can be inserted around line 174: "More than 75% of IDs from LRR-RLPs contain at least one of two lysine motifs, Kx5Y or Yx8KG, while less than 5% of IDs from LRR-RLK-Xb have either motif (Extended Figure 9b, Supplementary figure 33)." I feel that even just a short, motif level analysis of IDs would help match the comprehensive analysis of the C3 domain.

Response to reviewers: Many thanks for the suggestions. We are indeed unsure about the ancestry of LRR-RLK-Xb and LRR-RLPs, and it is unlikely that we can figure out whether LRR-RLK-Xb or LRR-RLP is the ancestral form. Nevertheless, we have restored the paragraph regarding the lysine motifs in the main text (yellow highlight). Furthermore, we have included statistical tests on the enrichment of both motifs in LRR-RLPs, and additional figures on the ID phylogenetic tree (Figure 6). We hope the readers will find these useful and get a clearer picture of the evolution of IDs.

2. Fig 4e demonstrates that RLP23 chimeras with BRI1 KD or TM-KD can induce BES1 de-phosphorylation. However, treatment with brassinolide (BL) as a positive control for BES1 bandshift is missing, and so the magnitude of de-phosphorylation (partial after 1 hr) cannot be compared to a full strength BRI1-mediated response. I expect BL would work through endogenous NbBRI1, but perhaps would require BRI1 overexpression. Brassinolide-induced responses are not normally studied in transient expression assays, and therefore I feel BL treatment is an important control not just for the current conclusion about RLP23-c-BRI1 function, but for establishing *N. benthamiana* transient expression as a system for RK signaling crosstalk.

Response to reviewers: Many thanks for the suggestions. We have repeated the assay with RLP23^{WT}, RLP23^{C-BRI1} and RLP23^{TM-BRI1} with nlp20 and BL treatment as a positive control (see supplementary figure 18). Indeed, BL does work through endogenous NbBRI1 and induces the de-phosphorylation of BES1 (without the need of BRI1 overexpression). The level of BES1 de-phosphorylation through NbBRI1 is comparable to the responses triggered by RLP23^{C-BRI1} and RLP23^{TM-BRI1} during nlp20 treatment. We hope these data will help to establish a *N. benthamiana* transient expression to study signal crosstalk in the future.

3. In data and materials availability, the authors state that "Sequences of the identified receptors and phylogenetic analyses will be uploaded and available to the public upon publication." This is a critical component for peer review. Please make these available through Zenodo or other repository as part of the revised manuscript.

Response to reviewers: Many thanks for the suggestions. We have uploaded the data associated with the manuscript to Zenodo at : <https://zenodo.org/records/10059978>

Minor points:

- Fig 1: It is not clear from the legend what the individual points represent on the expansion rate boxplots. Based on the text, my guess is a single species in the derived group vs a tge median percentage in the early diverging group? How many pairwise comparisons are represented in each panel? What is the x-axis unit?

Response to reviewers: Many thanks for the suggestions. We have included additional explanations in the figure legends and in the methods section.

- Fig 2+3: the term "PRR" is used in panel a, but throughout the text the authors are careful not to assume pattern recognition functions. I suggest removing "PRR" from the figures.

Response to reviewers: Many thanks for the suggestions. We have replaced 'PRRs' with 'cell-surface receptors' in the figures.

- Line 107 and Fig 3a: I can't tell if the black borders between pink and blue segments are thick dividing lines or separate RLK categories. Since only blue and pink sections are visible in the outer ring, how is this consistent with the "94.7%" claim in line 107?

Response to reviewers: Many thanks for the suggestions. We have removed the black borders in that figure. In addition, we have included the percentages of each category in the source data file.

- line 125: should refer to Extended figure 7, not 6

Response to reviewers: Many thanks. We have corrected this.

- line 164: "absence of a specific T residue before the Qxxx motif in RLP23" -- this phrasing makes it sound like RLP23 lacks the Thr. I suggest to just remove the words "in RLP23"

Response to reviewers: Many thanks for the suggestions. We have changed this sentence to "absence of a specific T residue before the Qxxx motif in **PSKR2 and BRI1**."

- Fig 4c: using 6-pointed Star of David (a religious symbol) for brassinosteroid is distracting. Is a 5-point star OK?

Response to reviewers: Many thanks for the suggestions. We have changed it into a 5-point star.

- The title feels overly broad, like a review rather than a primary data article. It is hard to tell from the current title how it differs from the previous analysis by Ngou et al 2022, Nature Plants, "Concerted expansion and contraction of immune receptor gene repertoires in plant genomes". I would suggest using the present title to draw attention to the RK-Xb and LRR-RLP findings.(something like "Functional conservation of LRR-associated regions between receptor kinases and receptor-like proteins")

Response to reviewers: Many thanks for the suggestions. After careful consideration, we have decided to retain our original title. We believe that it captures our study and findings better.

Reviewer #3 (Remarks to the Author):

The manuscript by Ngou et al. discussed the shared origin and function of cell surface receptors across the plant kingdom. In the first part of the manuscript, Ngou et al. expand on a previous curation of LRR cell surface receptors published last year to include additional ectodomains, and investigate the origin of the different receptor families along plant evolution, identifying the origin and expansion of different ectodomain-containing receptors in RLP and RLK forms. They find that the ectodomains themselves mostly exist already in Glaucophyta (except G-lectin and DUF26) and that the RLP form is usually ancestral to the RLK form. From that point, the paper focuses on LRR RLPs and RLKs and investigate their common origin, diverging roles and mechanistic basis for this divergence. At first, the authors focus on structural features in the LRR repeats, the C3, NL and ID motifs. They show that in LRR receptors containing IDs, RLPs and RLKs clade together in a phylogenetic analysis based on the C3 region, and conclude that these RLPs and RLKs share a common origin. They then show that a conserved motif in the C3 region is responsible for BAK coreceptor recruitment upon ligand binding using protein chimera based on RLP23. They then use the RLP23 LRR domain to generate chimeras in order to investigate the roles of the TM and surrounding regions in coreceptor recruitment and the roles of the kinase domain presence or absence in activation of developmental and immunological responses. The authors demonstrate that these chimeras are functional and define the protein regions responsible for recruitment and signaling, supporting the hypothesis that these various receptors share a common origin and underwent evolutionary diversification illustrated in Extended Data figure 10b to fulfill their diverse roles in immunity and development.

The paper presents a fascinating story of evolution, combines evolutionary analysis with functional protein assays to support the hypotheses, and includes multiple useful datasets. While the general concept of the interchangeability of ectodomains, signaling mechanisms and biological processes is not novel, the scale of the analysis and the elegant demonstration of the concepts are valuable, impressive and important for the advancement of our understanding of the topic. This paper is of great interest to the plant immunity field, but also to the fields of protein evolution and signal transduction. The analyses are in depth and comprehensive, and the results are generally well presented. I enjoyed reading the paper in depth, and believe many will find it interesting and useful.

Response to reviewers: We sincerely appreciate the valuable comments provided by the reviewer 3, which have greatly contributed to improving the quality of our manuscript.

Comments:

- One general criticism is that the paper is very long and contains a very large number of extended and supplementary data. Probably as a result of iterations in the revision process, the focus in the main text is not well reflected in the choice of the main figures. Main figure 2 is only superficially discussed in a single paragraph, while data in extended figure 1 for example is crucial for understanding but omitted from the main text. I would suggest rethinking what goes to the main figures and what goes to extended in light of the final version of the text presents. In hindsight, this work might have been more readable as two independent papers but this is of course at the discretion of the authors.

Response to reviewers: Many thanks for the suggestions. We did have some trouble fitting the large amount of data and ideas into one manuscript. While we are unable to split the paper into two, we have significantly rewritten the main text (see yellow highlights). We have also expanded the discussion on main figure 2

(now main figure 3). In addition, we have re-organized the main figures (now 9 main figures) according to the reviewers' suggestions. We hope the reviewers and readers will find this version more accessible and readable.

- Analysis of protein sequences derived from such a large number of genomes is prone to gene model annotation errors. This holds special importance as the analysis performed by the authors is searching for absence of domains and intervening sequences – both can be detected as a result of annotation errors of intron-exon junctions. While this is inherent to working with multiple complex eukaryotic genomes, some simple additional analyses could be added that will make the reader more confident.

- Such analyses should include: When only the ectodomain is detected, is it associated with another domain? Are those NLR genes? In genes annotated as RLPs, is there indeed only a short intracellular part to the protein or could they encode for an additional intracellular domain? Can this unannotated domain be a kinase, changing the picture regarding RLPs and RLKs? How many of the IDs and NLs detected are homologous to such sequences from close species?

- The use of a single PFAM annotation for kinase domain detection might cause false negative results for protein RLK detected as RLPs when the kinase is misannotated. Incorporation of additional kinase PFAM such as PF07714 or analysis of unannotated domains of receptors with more sensitive tools such as HHpred could support the results. As absence of a detected kinase domain is what makes a protein an RLP in this work, and as the claims regarding the evolutionary relation between RLPs and RLKs are central here, I believe a more careful analysis is merited to give the readers a more reliable picture. This can also be relevant to the discussion of the antiquity of the kinase domain at the end of paragraph about origin and expansion of cell-surface receptors.

Response to reviewers: Many thanks for the comments and suggestions. Please find below a list of additional analysis we have performed:

When only the ectodomain is detected, is it associated with another domain? Are those NLR genes?

- We have performed PFAM search on the domains that are associated with these ectodomain-only proteins (all seven types; please see supplementary data 1). For LRR ectodomain-only proteins, many of these are associated with pentatricopeptide repeats or DNA binding motifs (transcription factors). We do see NB-ARC domain enrichment, suggesting that at least a portion of them is indeed NLR genes.

In genes annotated as RLPs, is there indeed only a short intracellular part to the protein or could they encode for an additional intracellular domain? Can this unannotated domain be a kinase, changing the picture regarding RLPs and RLKs?

- We check the AA length of the intracellular part/domain in RLKs and RLPs (see supplementary data 1). While RLKs have much longer intracellular domains (around 300-400 AAs), RLPs mostly have very short intracellular domains (less than 100 AAs). Furthermore, we took the intracellular domains of the RLPs and performed PFAM search on them (see supplementary data 1). We failed to see any significant enrichment of any motifs, except for RING-finger domain in WAK-RLPs. We also do see protein kinase (PF07714.18) enriched in LysM-RLPs, but this only applies to a small number of protein and the match score is low (thus, likely to be non-functional kinase). To confirm that the RLPs are indeed not RLKs, we also ran additional search for kinase domains in RLPs (see below). Overall, we believe that the cytosolic regions in RLPs are mostly short and does not contain functional motifs or domains.

How many of the IDs and NLs detected are homologous to such sequences from close species?

- We took the LRR-RLP ID clusters/group and checked the total number of clusters with LRR-RLPs from species within one subclass (subclasses include Marchantiales, Anthocerotales, Sphagnales, Dicranales, Selaginellales, Salviniiales, Gnetales, Ginkgoales, Pinales, Amborellales, Nymphaeales, Laurales, Ranunculales, Acorales, Alismatales, Dioscoreales, Asparagales, Zingiberales, Arecales, Poales, Saxifragales, Caryophyllales, Cornales, Ericales, Apiales, Asterales, Gentianales, Lamiales, Solanales, Vitales, Myrtales, Malvales, Sapindales, Brassicales, Oxalidales, Fagales, Cucurbitales, Malpighiales, Fabales and Rosales). 99.62% of ID clusters contain LRR-RLPs from species within one subclass only (see supplementary data 2). Thus, the IDs detected are indeed closely related from close species, but not closely related at all from distant species from different subclasses. This also supports the idea that the IDs of LRR-RLPs have undergone extensive diversification following their divergence from LRR-RLK-Xbs.

The use of a single PFAM annotation for kinase domain detection might cause false negative results for protein RLK detected as RLPs when the kinase is misannotated. Incorporation of additional kinase PFAM such as PF07714 or analysis of unannotated domains of receptors with more sensitive tools such as HHpred could support the results.

- Thank you for the suggestion. We double-checked how many additional proteins would have been removed if we used the two additionally available PFAM patterns: PF07714.18 (Pkinase_Tyr) and PF00433.25 (Pkinase_C). We started with 10,224,245 proteins. The original search with PF00069.26 identified 479,270 proteins. Hence, 9,744,975 remained. If we add PF07714.18, we would identify an additional 1,506 proteins (0.015 %). If we also add PF00433.25 (protein kinase C), we would have identified another 104 proteins (0.002 %). Notably, the scores for both matches are mostly very poor, indicating that most are unlikely functional. Hence, the original search was relaxed enough to remove almost all proteins that might have any residual similarity to a protein kinase. Given that the effort to update everything is considerably high and the difference is very small, we would prefer to keep the filter using only the original PFAM pattern.

- The description of expansion and contraction of gene families in the main text is lacking, and is only well explained in the supplementary text. A clear explanation that % enrichment is in relation to the total number of genes in the genome (and not to the total number of PRRs for example) is required. Such analysis is also prone to artifacts as the expansion of another gene family in the genome would result in apparent contraction of the other gene families. This should be presented and discussed in the main text.

Response to reviewers: Many thanks for the suggestion. We have moved the supplementary text back into the main text. We have also included additional explanations in the methods section.

- “Moreover, downstream signalling components exhibit limited expansions compared to cell-surface receptors (Figure 2).” – what about the expansions of RLCK and CDPK? Are they considered non-significant?

Response to reviewers: Many thanks for the comments. As mentioned, we have expanded the discussion on main figure 2 (now main figure 3). Indeed, the RLCK and CDPK protein families have significantly expanded through the plant lineage, but not to the extent of cell-surface receptors (for example, LRR-RLKs) or NLRs. We have discussed possible causes of these expansions in the main text.

- Regarding Figure 3 - Are there results for co-IP of the PSKR2 chimera with the coreceptors? As the text refers to the inability of this chimera in inducing immune response and the importance of the T residue, these results are relevant.

Response to reviewers: Many thanks for the comments. We have included additional experiments with PSRK1, PSKR2, and point-mutant chimeras and co-IP results. The results imply that multiple residues both within and outside the TQxxT motif are important for SERK interactions. We concluded that the C3 region (including the last LRR motif) in LRR-RLPs and some LRR-RLK-Xbs (PSY1R and PSKR1) interact with SERKs in a similar manner, while some LRR-RLK-Xbs (PSKR2, BRI1) have likely evolved to interact with SERKs in a slightly different manner.

- The paragraph introducing the concepts of NLs and IDs is not referenced at all, preventing the reader from understanding what was known and what was discovered by this work. As a side note, the use of the acronym ID that is already used for “integrated domains” in NLR immune receptors is unfortunate, but not the fault of the authors.

Response to reviewers: Many thanks for the suggestions. We have referenced NLs and IDs with two reviews (<https://doi.org/10.3390/biom2020288>, <https://doi.org/10.1016/j.pmpp.2023.102004>).

- In the method section, it is not clear enough how datasets were generated – many references to “included” or “removed” when it is not clear enough what dataset was modified. Please specify exactly the proteomic dataset scanned, the search done, the number of hits identified, and refrain from basing the methods for this search on “as previously described”. Please provide final lists of protein accessions and sequences so the amazing dataset you created can be used by others, as beautifully done in previous works by the first author. Raw data for each of the trees and lists should be provided.

Response to reviewers: Many thanks for the suggestions. We have significantly rewritten the methods section and have uploaded the data associated with the manuscript to *Zenodo* at : <https://zenodo.org/records/10059978>

- References in the main text to Extended Data Figure 6 actually point to Ext. Data Fig. 7

Response to reviewers: Many thanks. We have corrected this.

- The graphical representation of the NL in Ext.Data Fig 5 is very hard to see. Highlighting it would be useful.

Response to reviewers: Many thanks for the suggestion. We have updated the figure and highlighted the IDs and NLs.

- While in the C3 phylogenetic tree in Fig. 3 it is clear that RLPs and RLKs are in sister clades, this is not so easily identifiable in the tree of Ext.Data Fig 9. Do the authors refer to the fact that the pink and blue leaves are intermixed in the same clades? It is hard to see visually. Better description of it in the text would be useful if the authors are making the claim that they are evolutionarily related.

Response to reviewers: Many thanks for the comments. We have added additional figures (main figure 6). The LRR-RLPs and LRR-RLK subgroups are now clearly labelled in the figure. It should be more obvious that the LRR-RLPs (pink) and LRR-RLK-Xbs (blue) are intermixed in the phylogenetic tree.

Reviewer #4 (Remarks to the Author):

The study presents a comprehensive comparative genomics analysis of RKL and RLP evolution by examining general multi-domain protein architectures. There are several interesting, focused analyses, including emergence and co-evolution of signaling co-receptors that is corroborated with molecular biology experiments. Additional comparative analyses are focused on motifs and their biochemical properties.

Response to reviewers: We sincerely appreciate the valuable comments provided by the reviewer 4, which have greatly contributed to improving the quality of our manuscript.

I think that the study would be greatly improved by statistical analyses, putting in context of other global RLK/RLP analyses as well as including structure-based interpretation of evolutionary and biochemical data. Please, see a list of suggestions and methodologies below:

- Protein family expansion and contraction are currently done with domain counting in organisms that are currently present. These are very general analyses that would be appropriate for a review but need to be done more robustly for a research paper to support presented conclusions. There are standard protein family evolutionary analyses quantifications that allow to model expansion and contraction based on comparing species phylogenies (single gene copy ortholog BUSCO genes) to protein family phylogenies. I think it would be essential to implement these. For suggested methods details, please see following paper for standard methodology:

o Shao ZQ, Xue JY, Wu P, Zhang YM, Wu Y, Hang YY, Wang B, Chen JQ. Large-Scale Analyses of Angiosperm Nucleotide-Binding Site-Leucine-Rich Repeat Genes Reveal Three Anciently Diverged Classes with Distinct Evolutionary Patterns. *Plant Physiol.* 2016 Apr;170(4):2095-109. doi: 10.1104/pp.15.01487. Epub 2016 Feb 2. PMID: 26839128; PMCID: PMC4825152.

o De Bie T, Cristianini N, Demuth JP, & Hahn MW (2006) CAFE: a computational tool for the study of gene family evolution. *Bioinformatics* 22(10):1269-1271.

o Hahn MW, De Bie T, Stajich JE, Nguyen C, & Cristianini N (2005) Estimating the tempo and mode of gene family evolution from comparative genomic data. *Genome research* 15(8):1153-1160.

Response to reviewers: Many thanks for the suggestion and the examples. We chose our approach because we have 350 plant genomes and multiple large gene families. Firstly, this means that the protein family trees contain a few thousand proteins just for a single domain/family. The papers mentioned were by no means close to this: Hahn et al. 2015 studied 5 yeast genomes. Shao et al. 2016 used 20 angiosperm species. However, they already had to resort to a step-by-step strategy. Secondly, there is no good phylogeny for all of our 350 species available. We originally tried to reconstruct a tree with tools like Orthofinder2, but that simply does not find any genes that could be used to reconstruct the species tree. We now searched all 350 genomes for single copy BUSCO genes and not a single one was found in all species. The maximum was a single gene in 300 species. Most of the BUSCO genes are found in about 250-270 species. We aligned each set separately (muscle), constructed trees (fasttree), generated a consensus tree that includes all species (ASTRAL), and converted it to an ultrametric tree with an arbitrary time scale using the function "chronos" from the R-package ape. We also tested CAFE5, but the tree and the gene families are simply too large. Also, it is not recommended to use the tool if the gene family is not already present in the root node (C. paradoxa). First, that would mean to use different trees, and second, it would also not resolve the problem

of absence in the basal lineages. Finally, run-times are extremely long and hardware requirements very high, we would again need to drop quite a few species and study only a subset. Hence, while it might be possible to do such an analysis on one or the other family, or maybe even on more if one would develop more code, it seems to be out of scope of the current study.

- There have been several previous publications on global RLK/RLP analyses that authors should take into account and compare results to:

o Man J, Gallagher JP, Bartlett M. Structural evolution drives diversification of the large LRR-RLK gene family. *New Phytol.* 2020 Jun;226(5):1492-1505. doi: 10.1111/nph.16455. Epub 2020 Feb 29. PMID: 31990988; PMCID: PMC7318236.

o Simon Snoeck Bradley W Abramson Anthony GK Garcia Ashley N Egan Todd P Michael Adam D Steinbrenner (2022) Evolutionary gain and loss of a plant pattern-recognition receptor for HAMP recognition *eLife* 11:e81050.

<https://doi.org/10.7554/eLife.81050>

Response to reviewers: Many thanks for the suggestion. We have incorporated and cited these two papers in our discussion.

- Motif enrichment analyses need to be statistically quantified, either by cumulative hypergeometric or cumulative binomial distributions.

Response to reviewers: Many thanks for the suggestion. The Fisher's exact test was performed to compare the number/fraction of IDs with either K5Y or Y8KG in LRR-RLPs against LRR-RLK-Xb. We confirmed that the K5Y and Y8KG motifs showed highly significant enrichment in LRR-RLPs (76.98%) compared to LRR-RLK-Xb (3.89%) ($p < 10e-16$; see figure 6c).

- There has been a breakthrough in structure modeling with AlphaFold2 so models for most genes in model species are available. Could you interpret evolutionary plus the biochemical data in structure context?

Response to reviewers: Many thanks for the suggestion. We have generated AF2 models for all the immune-related LRR-RLPs and checked their ID+C3 region (see supplementary figure 11). We have also generated additional chimeric LRR-RLPs to interpret the amino acid residues required for SERK interactions (see figure 5d-j).

Reviewer #1 (Remarks to the Author):

The authors have addressed most of my comments.

Reviewer #2 (Remarks to the Author):

My concerns have been addressed in the revision. I believe the manuscript represents a significant step forward for evolutionary origins of diverse-ectodomain RKs, and the functional relationship of subfamily Xb LRR-RK and LRR-RLPs. Both topics are will be excited to explore further for linking the evolution of immunity and development

Concerns in my review which were addressed:

- Multiple reviewers brought up motif enrichment in island domains and this has now been presented in lines 237-251, and the motif analysis has been moved to main Fig 6.
- A new experiment is presented in Suppl. Fig 18 which demonstrates that BES1 responses induced by chimeric RLPs are comparable to triggering endogenous BRI1. This supports conclusions about the activity of RLP23-BRI1 chimeras.
- Thanks for presenting alignments, lists, and trees on Zenodo

I would add in the context of other reviews that I find the C3 phylogenetic support for RK-Xb and RLP relationship to be convincing. I agree with another reviewer that the chimera functional data greatly support the hypothesis.

A minor comment: line 887 says "for better visualization, refer to Supplementary Fig. 15" but the figure was removed from the Supplement. With many reconfigurations of main and supplemental figures, a careful edit will be needed.

Reviewer #3 (Remarks to the Author):

My main concerns were adequately answered by the authors in their response. As I wrote in my initial review, I believe this paper will become a valuable resource for the field.

Minor comments:

- Typo in the legend of sup1 – mix up between x and y description
- In sup data 2 add total number of proteins so % will be understandable
- In Sup data 2 why did the LysM with PF07714 endodomain not included in the RLK list?
- In the legend of main fig 6, Reference to sup fig 15 should be to 16

Reviewer #4 (Remarks to the Author):

I appreciate the added rigor to the quantification and analyses and appreciate challenges associated with large datasets. While in theory calculating mean and looking at deviation is an ok method to compare gene sets, it suffers from not being able to incorporate evolutionary distance (and models associated with it), ancestral states as well as sequencing sampling bias and therefore should not have strong claim molecular evolutionary claims.

I agree with authors that there is room for improvement in large scale dataset analyses with evolutionary phylogenomics and I applaud them for trying to implement these analyses. I want to strongly caution interpretation of current quantification in "expansions" or "contractions" or calculating "rates of evolution rather than stating that these are comparisons of existing gene sets. For example, expansion rate needs to be calculated relative to the state in last common ancestors using evolutionary models and housekeeping genes, this is different from the presented analyses which compares gene sets, number of gene families members in organisms that live today. My

main suggestion at this point is to read through the manuscript and edit it carefully to reflect that current analyses compare "gene sets that exist today after substantial divergence and selection" rather than using molecular phylogenomics language which needs molecular phylogenomics quantification methods. I would suggest to add a few sentences in discussion about caveats of current data analyses, sequencing sampling biases and differences in organism lifestyles that we observe living today.

REVIEWERS' COMMENTS

Reviewer #1 (Remarks to the Author):

The authors have addressed most of my comments.

Response to reviewer: Thank you very much.

Reviewer #2 (Remarks to the Author):

My concerns have been addressed in the revision. I believe the manuscript represents a significant step forward for evolutionary origins of diverse-ectodomain RKs, and the functional relationship of subfamily Xb LRR-RK and LRR-RLPs. Both topics are will be excited to explore further for linking the evolution of immunity and development

Concerns in my review which were addressed:

- Multiple reviewers brought up motif enrichment in island domains and this has now been presented in lines 237-251, and the motif analysis has been moved to main Fig 6.
- A new experiment is presented in Suppl. Fig 18 which demonstrates that BES1 responses induced by chimeric RLPs are comparable to triggering endogenous BRI1. This supports conclusions about the activity of RLP23-BRI1 chimeras.
- Thanks for presenting alignments, lists, and trees on Zenodo

I would add in the context of other reviews that I find the C3 phylogenetic support for RK-Xb and RLP relationship to be convincing. I agree with another reviewer that the chimera functional data greatly support the hypothesis.

A minor comment: line 887 says "for better visualization, refer to Supplementary Fig. 15" but the figure was removed from the Supplement. With many reconfigurations of main and supplemental figures, a careful edit will be needed.

Response to reviewer: Thank you for the comments. We have checked and updated the main texts and figures.

Reviewer #3 (Remarks to the Author):

My main concerns were adequately answered by the authors in their response. As I wrote in my initial review, I believe this paper will become a valuable resource for the field.

Minor comments:

- Typo in the legend of sup1 – mix up between x and y description
- In sup data 2 add total number of proteins so % will be understandable
- In Sup data 2 why did the LysM with PF07714 endodomain not included in the RLK list?
- In the legend of main fig 6, Reference to sup fig 15 should be to 16

Response to reviewer: Thank you for the comments.

-Typo in the legend of supp data 2 (now 1a): We have fixed this.

-In sup data 2 (now 1b), total number of proteins (n) are added to the list.

-For Sup data 2 (now 1b): during the initial LysM-RLK search in Ngou et al, *Nature Plants* 2022, PF07714.18 was not used to search for a kinase domain in LysM-RLKs. Thus, some proteins from the LysM-RLP list contain endo-domain with PF07714.18 matches. These 53 LysM-RLPs might be potential LysM-RLKs. We have stated that in the legend of Sup data 1b. We have also included a list of these 53 potential LysM-RLKs in the legend.

-We have removed the reference to supp figure 15.

Reviewer #4 (Remarks to the Author):

I appreciate the added rigor to the quantification and analyses and appreciate challenges associated with large datasets. While in theory calculating mean and looking at deviation is an ok method to compare gene sets, it suffers from not being able to incorporate evolutionary distance (and models associated with it), ancestral states as well as sequencing sampling bias and therefore should not have strong claim molecular evolutionary claims.

I agree with authors that there is room for improvement in large scale dataset analyses with evolutionary phylogenomics and I applaud them for trying to implement these analyses. I want to strongly caution interpretation of current quantification in "expansions" or "contractions" or calculating "rates of evolution" rather than stating that these are comparisons of existing gene sets. For example, expansion rate needs to be calculated relative to the state in last common ancestors using evolutionary models and housekeeping genes, this is different from the presented analyses which compares gene sets, number of gene families members in organisms that live today. My main suggestion at this point is to read through the manuscript and edit it carefully to reflect that current analyses compare "gene sets that exist today after substantial divergence and selection" rather than using molecular phylogenomics language which needs molecular phylogenomics quantification methods. I would suggest to add a few sentences in discussion about caveats of current data analyses, sequencing sampling biases and differences in organism lifestyles that we observe living today.

Response to reviewer: Thank you for the comments. We agree with Reviewer 4. We have added a section in Supplementary Note 2 on the limitation of our current analysis.